# GENERALIZABLE PERSON RE-IDENTIFICATION WITHOUT DEMOGRAPHICS

## ABSTRACT

Domain generalizable person re-identification (DG-ReID) aims to learn a ready-to-use domain-agnostic model directly for cross-dataset/domain evaluation, while current methods mainly explore the demographic information such as domain and/or camera labels for domain-invariant representation learning. However, the above-mentioned demographic information is not always accessible in practice due to privacy and security issues. In this paper, we consider the problem of person re-identification in a more general setting, *i.e.,* domain generalizable person re-identification without demographics (**DGWD-ReID**). To address the underlying uncertainty of domain distribution, we introduce distributionally robust optimization (DRO) to learn robust person re-identification models that perform well on all possible data distributions within the uncertainty set without demographics. However, directly applying the popular Kullback-Leibler divergence constrained DRO (or KL-DRO) fails to generalize well under the distribution shifts in real-world scenarios, since the convex condition may not hold for overparameterized neural networks. Inspired by this, we analyze and reformulate the popular KL-DRO by applying the change-of-measure technique, and then propose a simple yet efficient approach, **Unit-DRO**, which minimizes the loss over a new dataset with hard samples up-weighted and other samples down-weighted. We perform extensive experiments on both domain generalizable and cross-domain person ReID tasks, and the empirical results show that Unit-DRO achieves superior performance compared to all baselines without using demographics.

## 1 INTRODUCTION

Person re-identification (ReID) aims to find the correspondences between person images from the same identity across multiple camera views. As illustrated in Figure 1, previous studies mainly follow three different settings: 1) **supervised person ReID** Zhang et al. (2020), where training and test data are independently and identically (*i.i.d*) drawn from the same distribution. Though recent supervised methods have achieved remarkable performance, they are usually non-robust in out-of-distribution (OOD) settings; 2) **unsupervised domain adaptive person ReID (UDA-ReID) and cross-domain person ReID (CD-ReID)** Luo et al. (2020), where UDA-ReID relies on large amounts of unlabeled data for retraining and CD-ReID cannot exploit the benefits brought by multi-source domains; 3) **domain generalizable person ReID (DG-ReID)** Dai et al. (2021a), where the model is trained on multiple large-scale datasets and tested on unseen domains directly without extra data collection/annotation and model updating on new domains. Therefore, DG-ReID is receiving increasing attention due to its great value in real-world person retrieval applications.

However, current DG-ReID research usually comes at a serious disadvantage: it requires the demographic information (*e.g.,* domain labels Choi et al. (2021); Zhao et al. (2021), camera IDs Zhang et al. (2021b); Dai et al. (2021a), and video timestamps Yuan et al. (2020)) as the extra supervision for model training. Such demographics implicitly define the variations in training data that the learned model should be invariant or robust to. Unfortunately, the demographic information is usually not available in practice due to the following reasons: 1) the collection of demographics inevitably leads to privacy problems Veale & Binns (2017), *e.g.,* the risks of exposing the geographical location and/or the environment information; 2) the collection/annotation of domain labels is very expensive and ethically fraught endeavours Michel et al. (2021); and 3) such coarse-grained labels and the noise of manual annotation collected domain labels may exacerbate the *hidden stratification issue*, which hinders a variety of safety-critical applications Creager et al. (2021); Kim & Lee (2021) (see Appendix A for more discussions). Therefore, as shown in Figure 1d, we consider a more general

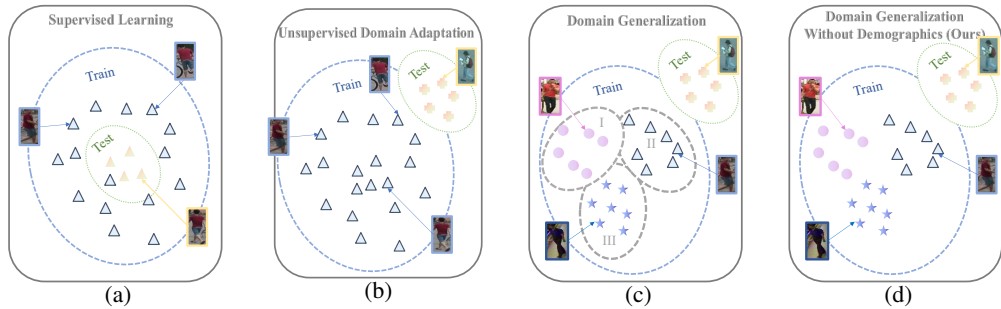

Figure 1: An illustration of different person re-identification settings. (a) Supervised person ReID. (b) CD-ReID and UDA-ReID. (c) DG-ReID. (d) DGWD-ReID.

setting for person ReID, *i.e.,* Domain Generalizable Person Re-identification Without Demographics (DGWD-ReID), where the model is trained on multiple large-scale datasets *without demographics*.

To address the underlying uncertainty of domain distribution without using demographics, distributionally robust optimization (DRO) is a promising paradigm, which explicitly obtains prediction functions robust to distribution shifts Hu et al. (2018). Specifically, DRO considers a minimax game: the inner optimization objective is to shift the training distribution within a pre-specified uncertainty set so as to maximize the expected loss on the test distribution. The outer optimization minimizes the adversarial expected loss. The uncertainty set defined by an $f$-divergence ball (such as Kullback-Leibler divergence) from the training distribution has been very popular, which is also known as KL-DRO Hu & Hong (2013). However, the convex assumption in KL-DRO usually cannot hold in real-world scenarios, thus leading to inferior performance for overparameterized neural networks.

We address the above-mentioned issue and reformulate KL-DRO to first solve the inner step optimization problem and then obtain a closed-form expression of the optimal objective. Specifically, different from previous work that converts the minimax DRO problem into a single minimization problem by the closed-form expression Hu & Hong (2013), we utilize a change-of-measure technique and reformulate the minimax optimization as an importance sampling problem, termed Unit-DRO[1]. By doing this, Unit-DRO avoids bi-level optimization in traditional DRO problems and scales well to overparameterized regimes. Specifically, Unit-DRO upweights samples that are prone to be misclassified and downweights others. It assigns a normalized weight $e^{\ell/\tau^*}/\mathbb{E}[e^{\ell/\tau^*}]$ to each pair of data and label $(x, y)$, where $\ell$ indicates the error incurred by $(x, y)$ and $\tau^*$ is a hyperparameter. During implementation, there are still two main challenges for applying Unit-DRO: 1) it struggles with the hyperparameter parameter $\tau^*$ and we observe that a constant $\tau^*$ during training always leads to inferior performance in practice; 2) the normalization factor $\mathbb{E}[e^{\ell/\tau^*}]$ requires an expectation over the training distribution, which is not complementary with the stochastic mini-batch training. To tackle the first problem, we propose a **multi-step** solution to adaptively determine the value of $\tau^*$ by the training step. We then maintain a **weight queue** to store historical sample weights for a better estimation of $\mathbb{E}[e^{\ell/\tau^*}]$ over the training distribution. Compared to previous DG-ReID methods, Unit-DRO avoids the need for either meta-learning pipelines or model structure engineering.

In this paper, we evaluate the proposed Unit-DRO for person ReID by comparing it with existing DG-ReID and CD-ReID methods. Unit-DRO outperforms a variety of recent methods with a large margin on both DG-ReID and CD-ReID benchmarks, even including those methods using demographics. To better understand the proposed Unit-DRO, we perform comprehensive ablation studies on several important components, such as the multi-step $\tau^*$ solution and the weight queue. Furthermore, we also visualize the learned weight distributions, $t$-SNE embeddings, and measure the domain divergence and error set to show the good invariant learning capability of Unit-DRO. Empirical results show that the proposed Unit-DRO can effectively retrieve valuable samples or subgroups without demographics.

## 2    RELATED WORK

**DG-ReID**. Generalizable methods are recently proposed to learn invariant representations that can generalize to unseen domains Song et al. (2019); Choi et al. (2021); Zhang et al. (2021b).

---

[1]In contrast to the word "Group" in Group-DRO Sagawa et al. (2019) where it assigns weights for domains, the word "Unit" in our proposed Unit-DRO assigns weights for samples.

Existing methods mainly utilize domain divergence minimization strategies or a meta-learning pipeline. In view of the current research trend (Table 1), most methods rely on demographics to learn invariant features. Though existing strong baseline Liao & Shao (2020), normalization Jin et al. (2020), and augmentation methods Yan et al. (2020) require no demographics, they are plug-and-play modules and thus orthogonal to the proposed Unit-DRO. Different from existing studies, DGWD-ReID adds a strict restriction on demographics and has ambitious targets that **"can we learn invariant features even without demographics? can we partition domains better?"**.

**Fairness without Demographics.** Methods in Fairness Dwork et al. (2012) aim to develop a model that performs well for worst-case group assignments according to some fairness criteria for addressing the underperformance in minority subgroups. Though there are several works considering *fariness without demographics* Liu et al. (2021); Creager et al. (2021), they mostly evaluate their algorithms in datasets with predefined distribution shifts. Note that DGWD-ReID is more challenging than the category-level recognition problem considered in the existing *fairness w or w/o demographics* study. In DGWD-ReID, the target identities are different from source ones and we need to tackle both domain gap and disjoint label space problems simultaneously. For more discussions about domain generalization, DRO, and cross-domain person ReID (CD-ReID), please also refer to Appendix B.

Table 1: The current research trend of DG-ReID.

| Method | Source | Domain | Camera |
|---|---|---|---|
| DIR-ReID Zhang et al. (2021b) | Arxiv 21 | ✓ | ✓ |
| MetaBIN Choi et al. (2021) | CVPR 21 | ✓ | ✓ |
| M3L Zhao et al. (2021) | CVPR 21 | ✓ | |
| DMG-Net Bai et al. (2021) | CVPR 21 | ✓ | ✓ |
| RaMoE Dai et al. (2021b) | CVPR 21 | ✓ | |
| CBN Zhuang et al. (2020) | ECCV 20 | | ✓ |
| CAIL Luo et al. (2020) | ECCV 20 | ✓ | ✓ |
| QAConv Liao & Shao (2019) | ECCV 20 | Backbone | |
| SNR Jin et al. (2020) | CVPR 20 | Normalization | |

## 3 METHOD

**Problem Formulation.** Given the current DG-ReID setting, there is a labeled set of training data from several different domains: $\mathcal{P} = \cup_{k=1}^N P_k$ and $P_k = \{(x_i, y_i)\}_{i=1}^{N_k}$, where $N$ is the number of domains, $N_k$ is the number of images in domain $P_k$, and $x_i \in \mathcal{X}, y_i \in \mathcal{Y}$ indicate an image and its corresponding label, respectively. During training, we use all aggregated image-label pairs from $\mathcal{P}$. During testing, we evaluate the person retrieval performance on the unseen target domain $G$ without any additional model updating. Therefore, the goal of DG-ReID is to learn a model $f_\theta : \mathcal{X} \rightarrow \mathcal{Y}$ that minimizes the empirical error on the unseen target domain $G$:

$$\min_{\theta \in \Theta} \mathbb{E}_{(x,y) \in G} \left[ \ell(x, y; \theta) \right], \tag{1}$$

where $\ell$ is the predefined loss function. This objective encodes the goal of learning a model that does not depend on spurious correlations. If a model makes decisions according to domain-specific information, it is natural to be brittle in an entirely distinct domain. However, previous studies mostly leverage demographics (*e.g.,* domain/camera labels and video timestamps) to clip the spurious correlations, which is not always available in real-world applications. Therefore, we consider a more general setting where the above-mentioned demographic information is unknown during training, *i.e.,* DG-ReID without demographics or DGWD-ReID, which is in line with the motivation that annotating demographics is expensive and also likely to expose privacy information.

**Baseline Algorithm.** We introduce the objectives used in our baseline as follows. The first one is the cross-entropy loss. Given $n$ training points $\{(x_1, y_1), ..., (x_n, y_n)\}$, we then have the loss for person identity classification: $\mathcal{L}_{ce} = \frac{1}{n} \sum_{i=1}^n \ell(x_i, y_i; \theta)$, where $\ell$ indicates the cross-entropy loss function. Label-smoothing is also applied to prevent the model from overfitting to the identity labels. Inspired by recent ReID methods, we further introduce triplet loss to enhance the intra-class compactness and inter-class separability in the embedding space. Following Hermans et al. (2017), given an anchor sample $x_i^a$, we then evaluate triplet loss using the hardest positive and negative samples, $x_i^p$ and $x_i^n$ within each mini-batch: $\mathcal{L}_{tr}(x_i^a, x_i^p, x_i^n; \theta) = \max \{d(x_i^a, x_i^p; \theta) - d(x_i^a, x_i^n; \theta) + m, \, 0\}$, where $d(\cdot, \cdot)$ indicates a pairwise distance such as the Euclidean distance, and $m$ is the margin between positive and negative pairs. We use a BNNeck structure Luo et al. (2019a) to maximize the synergy between $\mathcal{L}_{ce}$ and $\mathcal{L}_{tr}$ and integrate a mixture of batch normalization and instance normalization with learnable parameters Choi et al. (2021), which are shown very useful for DG-ReID. In the following, we reuse $\ell(x, y; \theta)$ as the sum of both cross-entropy loss and triplet loss.

### 3.1 UNIT-DRO

To address the underlying uncertainty of domain distribution without demographics, we introduce Unit-DRO, a novel generalization framework that does not require priors about demographics. We first introduce the basic distributionally robust optimization (DRO) framework Ben-Tal et al. (2009); Rahimian & Mehrotra (2019) as follows. In DRO, the worst-case expected risk over a predefined family of distributions $\mathcal{Q}$ (termed *uncertainty set*) is used to replace the expected risk on the unseen target distribution $G$ in Equ.1. Therefore, the objective is as follows,

$$\min_{\theta \in \Theta} \max_{q \in \mathcal{Q}} \mathbb{E}_{(x,y) \in q}[\ell(x, y; \theta)]. \tag{2}$$

Specifically, the uncertainty set $\mathcal{Q}$ encodes the possible test distributions that we want our model to perform well on. If $\mathcal{Q}$ contains $G$, the DRO object can upper bound the expected risk under $G$.

An important question for using DRO is how to choose the uncertainty set (please also see more discussions in Appendix. B). Note that in real-world applications, we can obtain only the empirical (training) data distribution. The uncertainty set can thus be constructed by collecting the distributions within a certain distance from the training distribution. For example, previous work may choose a KL-divergence ball Hu & Hong (2013)/MMD ball Sinha et al. (2017) around the training distribution, which confers robustness to a wide set of distribution shifts. However, it can also lead to overly pessimistic models which optimize for implausible worst-case distributions Duchi et al. (2019). In other words, $\mathcal{Q}$ should be sufficiently large to contain $G$, while it may also contain noisy distributions Michel et al. (2021). Group-DRO Sagawa et al. (2019) thus leverages demographics to define the uncertainty set $\mathcal{Q}$ and attains superior OOD performance. Different from Group-DRO, here we consider a new extension of DRO to improve OOD generalization without demographics.

**KL-DRO**. We first introduce the construction of uncertainty set $\mathcal{Q}$ based on the KL-divergence ball around the empirical distribution $\mathcal{P}$. Given the KL upper bound (radius) $\eta$, we have the uncertainty set $\mathcal{Q} = \{Q : \text{KL}(Q||\mathcal{P}) \leq \eta\}$. The min-max problem in Equ.2 can then be reformulated as

$$\min_{\theta \in \Theta} \max_{Q: \text{KL}(Q||\mathcal{P}) \leq \eta} \mathbb{E}_{(x,y) \in Q}\left[\ell(x, y; \theta)\right]. \tag{3}$$

**Lemma 1** *(Modified from Section 2 in Hu & Hong (2013)) Assume the model family $\theta \in \Theta$ and $\mathcal{Q}$ to be convex and compact. The loss $\ell$ is continuous and convex for all $x \in \mathcal{X}, y \in \mathcal{Y}$. Suppose empirical distribution $\mathcal{P}$ has density $p(x, y)$. Then the inner maximum of Equ.3 has a closed-form solution*

$$q^*(x, y) = \frac{p(x, y)e^{\ell(x,y;\theta)/\tau^*}}{\mathbb{E}_{\mathcal{P}}\left[e^{\ell(x,y;\theta)/\tau^*}\right]}, \tag{4}$$

*where $\tau^*$ satisfies $\mathbb{E}_{\mathcal{P}}\left[\frac{e^{\ell(x,y;\theta)/\tau^*}}{\mathbb{E}_{\mathcal{P}}[e^{\ell(x,y;\theta)/\tau^*}]}\left(\frac{\ell(x,y;\theta)}{\tau^*} - \log \mathbb{E}_{\mathcal{P}}[e^{\ell(x,y;\theta)/\tau^*}]\right)\right] = \eta$ and $q^*(x, y)$ is the optimal density of $Q$. The min-max problem in Equ.3 is then equivalent to*

$$\min_{\theta \in \Theta, \tau > 0} \tau \log \mathbb{E}_{\mathcal{P}}\left[e^{\ell(x,y;\theta)/\tau}\right] + \eta\tau. \tag{5}$$

We refer to Equ.5 as **KL-DRO**. Unfortunately, the convex condition of KL-DRO is not held for overparameterized neural networks, such that applying it may fail to generalize under the distribution shifts in real-world scenarios. As illustrated in Figure 2, we compare the training statistics with the baseline, where KL-DRO is highly unstable and attains inferior results. Therefore, instead of following KL-DRO to directly use the inner maximum, we reformulate Equ.3 as follows.

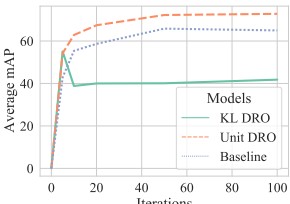

Figure 2: Training statistics.

$$
\begin{aligned}
\min_{\theta \in \Theta} \max_{Q: \text{KL}(Q||\mathcal{P}) \leq \eta} \mathbb{E}_{(x,y) \in Q}[\ell(x, y; \theta)] &= \min_{\theta \in \Theta} \max_{Q: \text{KL}(Q||\mathcal{P}) \leq \eta} \int \ell(x, y; \theta) q(x, y) d_x d_y \\
&= \min_{\theta \in \Theta} \max_{Q: \text{KL}(Q||\mathcal{P}) \leq \eta} \int \ell(x, y; \theta) \frac{q(x, y)}{p(x, y)} p(x, y) d_x d_y \\
&= \min_{\theta \in \Theta} \max_{Q: \text{KL}(Q||\mathcal{P}) \leq \eta} \mathbb{E}_{(x,y) \in \mathcal{P}}\left[\frac{q(x, y)}{p(x, y)} \ell(x, y; \theta)\right] \\
&= \min_{\theta \in \Theta} \mathbb{E}_{(x,y) \in \mathcal{P}}\left[\frac{e^{\ell(x,y;\theta)/\tau^*}}{\mathbb{E}_{\mathcal{P}}[e^{\ell(x,y;\theta)/\tau^*}]} \ell(x, y; \theta)\right].
\end{aligned}
\tag{6}
$$

Specifically, to obtain the third line, we apply the change-of-measure technique. The fourth line replaces the inner maximum by its closed-form solution $q^*(x, y)$ in Equ.4. Note that both the value of $\tau^*$ and the normalizer $\mathbb{E}_{\mathcal{P}}[e^{\ell(x,y;\theta)/\tau^*}]$ depend on the expectation of losses over all training data, which is untrackable at each mini-batch based optimization step. For simplicity, we can serve $\tau^*$ as a hyperparameter and take the average over each mini-batch as a preliminary estimator of the normalizer. Therefore, we have the formulation of vanilla Unit-DRO as follows:

$$\mathcal{L}_{\text{Unit-DRO}}(\theta, \tau^*) = \min_{\theta \in \Theta} \frac{1}{N} \sum_{i=1}^{N} \left( \frac{e^{\ell(x,y;\theta)/\tau^*}}{\frac{1}{N} \sum_{i=1}^{N} \left( e^{\ell(x,y;\theta)/\tau^*} \right)} \ell(x, y; \theta) \right), \tag{7}$$

where $N$ is the batch size. However, vanilla Unit-DRO does not work well in practice, and we address the following two problems to form a robust Unit-DRO solution.

**Multi-Step $\tau^*$.** The first problem is that a constant hyperparameter $\tau^*$ is usually suboptimal for the whole learning process. As shown in Figure 3, we visualize the densities of the weight $e^{\ell(x,y;\theta)/\tau^*}/\mathbb{E}_{\mathcal{P}}[e^{\ell(x,y;\theta)/\tau^*}]$ at different optimization steps when using a constant $\tau^*$ (please refer to Section 4.3 for the detailed setups). Specifically, we find that: 1) a small $\tau^*$ leads to the high variance on the weight distribution and is also sensitive to outliers; 2) a large $\tau^*$ is so conservative that the weights for all samples are almost similar, and the performance is thus similar to the baseline method. To tackle this problem, we propose a multi-step solution for the hyparameter $\tau^*$, which declines with the training/optimization steps. The intuition behind the multi-step $\tau^*$ is that: at the beginning, we use a large $\tau^*$, and the model thus assigns almost similar weights to all samples and cannot identify which sample is more important or not. With the increase of training steps, we decrease the value of $\tau^*$ and improve the weights for important (*i.e.,* hard-to-distinguish) samples.



Figure 3: Visualizing the distribution of sample weight at $1k, 5k, 10k, 20k$ steps, respectively (from left to right). The horizontal axis represents the weight.

**Weight Queue $\mathcal{M}$.** The second problem is that the expectation over each mini-batch may not be a good estimator of the normalizer $\mathbb{E}_{\mathcal{P}}[e^{\ell(x,y;\theta)/\tau^*}]$. To address this problem, we introduce a queue $\mathcal{M} = \{w_i := e^{\ell(x_i,y_i;\theta)/\tau^*}\}_{i=1}^{M}$ to maintain the historical weights, where $M$ depends on the batch size $N$ and determines how well $\mathcal{M}$ can estimate $\mathbb{E}_{\mathcal{P}}[e^{\ell(x,y;\theta)/\tau^*}]$. (Please see more empirical analysis in Section 4.3).

Lastly, we have the objective function of **Unit-DRO** as follows:

$$\mathcal{L}_{\text{Unit-DRO}}(\theta, \tau^*(t)) = \min_{\theta \in \Theta} \frac{1}{N} \sum_{i=1}^{N} \left( \frac{e^{\ell(x,y;\theta)/\tau^*(t)}}{\frac{1}{|\mathcal{M}|} \sum_{w_i \in \mathcal{M}} (w_i)} \ell(x, y; \theta) \right), \tag{8}$$

where $t$ is the index of training step and $\tau^*$ is a piecewise function of $t$. As shown in Figure 2, the training statistics of Unit-DRO is more stable than KL-DRO, and its performance also outperforms baseline methods by a large margin. We depict the online optimization algorithm in Appendix Algorithm 1. Note that in Algorithm 1 of Group-DRO Sagawa et al. (2019), all samples in the same domain share the same weight, which can be seen as a special case of the proposed Unit-DRO. Compared with Group-DRO, one of the key improvements is the implementation trick in that the group weights are updated using exponential gradient ascent instead of picking the group with the worst average loss at each step. Specifically, Group-DRO shows that such an improvement is useful for training stability and model convergence but cannot explain why it works. In contrast, the adaptive weights used in this paper are interpretable: the optimal distribution of DRO with KL constraint is proportional to the empirical distribution composite with the exponential term $e^{\ell(x,y;\theta)/\tau^*}$.

## 4 EXPERIMENTS

In this section, we evaluate the proposed Unit-DRO and try to answer the following questions: "without demographics, how does Unit-DRO perform compared to other CD-ReID and DG-ReID methods? what is the influence of different hyperparameters in Unit-DRO? why Unit-DRO improves the baseline?". To answer the first question, we compare Unit-DRO with baseline methods on both DG-ReID and CD-ReID benchmarks. We then perform detailed ablation studies to answer the second question. Comprehensive analyses are conducted for the third question, *e.g.,* error set analysis, feature visualization, and domain divergence measure.

### 4.1 EXPERIMENTAL SETUP

**Datasets.** Following Song et al. (2019); Jia et al. (2019); Zhang et al. (2021b), we evaluate the Unit-DRO with multiple data sources (MS), where source domains cover five large-scale ReID datasets, including CUHK02 Li & Wang (2013), CUHK03 Li et al. (2014), Market1501 Zheng et al. (2015), DukeMTMC-ReID Zheng et al. (2017), and CUHK-SYSU PersonSearch Xiao et al. (2016). The unseen test domains are VIPeR Gray et al. (2007), PRID Hirzer et al. (2011), QMUL GRID Liu et al. (2012), and i-LIDS Wei-Shi et al. (2009). We include the detailed illustration of datasets and evaluation protocols in Appendix D.1. In the CD domain setting, we employ Market1501 and DukeMTMC-ReID. We alternately construct the two datasets into source and target domains.

**Baselines** We compare our model with 1) **DG-ReID** methods, including AugMining Tamura & Murakami (2019), DIMN Song et al. (2019), DualNorm Jia et al. (2019), SNR Jin et al. (2020), DDAN Chen et al. (2021), DIR-ReID Zhang et al. (2021b), and MetaBIN Choi et al. (2021); and 2) **CD-ReID** methods, including Cross-Grad Shankar et al. (2018), QAConv Liao & Shao (2019), L2A-OT Zhou et al. (2020), OSNet-AIN Zhou et al. (2021), SNR Jin et al. (2020), DIR-ReID Zhang et al. (2021b), and MetaBIN Choi et al. (2021).

Table 2: Summary of different DG-ReID protocols. (M:market1501, C2: Cuhk02, C3: Cuhk03, D: DukeMTMC, MT: MSMT17, CS: CUHK-SYSU, V: ViPeR, P: PRID, G: GRID, I: i-LIDS).

| Protocol | Source | Target | Augmentation |
|---|---|---|---|
| (1) | M/D | D/M+V+P+G+I | Color-Jittering |
| (2) | MS+D+M (train) | C3 | None |
| (3) | M+D+MT | C3 | Color-Jittering |
| (4) | M+D+C3+MT | V+P+G+I | Color-Jittering |

**Implementation Details.** Following previous DG-ReID methods, we use MobileNetV2 Sandler et al. (2018) with the width multiplier of $1.4$ as the backbone network, which is initialized using the weights pre-trained on ImageNet Deng et al. (2009). All training images are resized to $256 \times 128$ pixels and the batch size is $N = 80$. We use the SGD optimizer with momentum $0.9$ and the weight decay $5e - 4$. The learning rate starts from $0.01$ and then decays to its $0.1\times$ at 40 and 70 epochs. We also use a warmup learning rate schedule at the first 10 epochs. We initialize the multi-step $\tau^*$ with $\tau^* = 100$, which is then decayed to 20 and 5 at 40 and 70 epochs, respectively. The default size of the weight queue is $M = 800$. The training process includes 100 epochs. During training, we also use the label-smoothing with the parameter $0.1$ and the margin of triplet loss is $0.3$. We conduct all the experiments on a machine with i7-8700K CPU, 32G RAM, and four GeForce RTX2080Ti (12GB) GPU cards.

### 4.2 RESULTS

**DG-ReID**. Considering that most existing methods can not work without demographic information, we thus compare the proposed Unit-DRO with methods on a typical DG-ReID setting, *i.e.,* all other methods use the demographics except Unit-DRO. As shown in Table 3, the proposed Unit-DRO significantly outperforms Group-DRO, while it also achieves either comparable or better performance when compared with recent state-of-the-art DG-ReID methods using demographics. By doing this, we hope that the proposed Unit-DRO can serve as a strong baseline for both DG-ReID and DGWD-ReID. We observe that current DG ReID methods all apply a utopian model selection method to **report their best result by carefully checking the test performance after each training epoch**. So the numbers of training epochs corresponding to the best performance are varying for different test datasets We argue that such a model selection method is inadvisable. Under the DG setting, we should restrict access to the test domain data for model selection. Thus, we use the last checkpoint and report its results as the final performance over all test datasets. The results in Table 3 show that, without the utopian model selection method, there is always a certain degree of performance decline for existing DG ReID methods, which further indicates the advantages of the proposed Unit-DRO..

Table 3: Comparison with recent state-of-the-art DG-ReID methods. † means the results of the last checkpoint are reported.

| Methods | Average | | VIPeR | | | | PRID | | | | GRID | | | | i-LIDS | | | |
|---|---|---|---|---|---|---|---|---|---|---|---|---|---|---|---|---|---|---|
| | R-1 | mAP | R-1 | R-5 | R-10 | mAP | R-1 | R-5 | R-10 | mAP | R-1 | R-5 | R-10 | mAP | R-1 | R-5 | R-10 | mAP |
| AugMining | 51.8 | - | 49.8 | 70.8 | 77.0 | - | 34.3 | 56.2 | 65.7 | - | 46.6 | 67.5 | 76.1 | - | 76.3 | 93.0 | 95.3 | - |
| DIMN | 47.5 | 57.9 | 51.2 | 70.2 | 76.0 | 60.1 | 39.2 | 67.0 | 76.7 | 52.0 | 29.3 | 53.3 | 65.8 | 41.1 | 70.2 | 89.7 | 94.5 | 78.4 |
| DualNorm | 57.6 | 61.8 | 53.9 | 62.5 | 75.3 | 58.0 | 60.4 | 73.6 | 84.8 | 64.9 | 41.4 | 47.4 | 64.7 | 45.7 | 74.8 | 82.0 | 91.5 | 78.5 |
| DDAN | 59.0 | 63.1 | 52.3 | 60.6 | 71.8 | 56.4 | 54.5 | 62.7 | 74.9 | 58.9 | 50.6 | 62.1 | 73.8 | 55.7 | 78.5 | 85.3 | 92.5 | 81.5 |
| DDAN w/DualNorm | 60.9 | 65.1 | 56.5 | 65.6 | 76.3 | 60.8 | 62.9 | 74.2 | 85.3 | 67.5 | 46.2 | 55.4 | 68.0 | 50.9 | 78.0 | 85.7 | 93.2 | 81.2 |
| DIR-ReID | 63.8 | 71.2 | 58.5 | 76.9 | 83.3 | 67.0 | 69.7 | 85.8 | 91.0 | 77.1 | 48.2 | 67.1 | 76.3 | 57.6 | 79.0 | 94.8 | 97.2 | 83.4 |
| DIR-ReID† | 62.3 | 70.8 | 57.2 | 74.1 | 80.2 | 64.9 | 67.6 | 87.1 | 91.6 | 76.6 | 47.2 | 66.1 | 75.4 | 57.0 | 77.3 | 93.3 | 97.2 | 84.5 |
| MetaBIN | 64.7 | 72.3 | 56.9 | 76.7 | 82.0 | 66.9 | 72.5 | 88.2 | 91.3 | 79.8 | 49.7 | 67.5 | 76.8 | 58.1 | 79.7 | 93.3 | 97.0 | 85.5 |
| MetaBIN† | 64.2 | 71.9 | 59.3 | 76.8 | 81.9 | 67.6 | 70.6 | 86.5 | 91.5 | 78.2 | 47.3 | 66.0 | 74.0 | 56.4 | 79.5 | 93.0 | 97.5 | 85.5 |
| Group-DRO | 57.1 | 65.9 | 48.5 | 68.4 | 77.2 | 57.8 | 66.1 | 86.5 | 90.6 | 74.8 | 38.7 | 58.8 | 66.6 | 48.6 | 74.8 | 90.8 | 96.8 | 81.9 |
| Group DRO† | 56.7 | 65.6 | 48.5 | 68.9 | 76.6 | 58.1 | 65.4 | 85.4 | 89.8 | 74.1 | 38.4 | 58.6 | 66.1 | 48.4 | 74.5 | 91.0 | 96.0 | 81.7 |
| **Unit-DRO†** | **65.4** | **72.8** | **60.0** | **78.2** | 82.8 | **68.4** | **73.5** | 85.3 | **91.7** | 79.4 | 47.5 | **69.3** | **77.4** | 57.2 | **80.7** | 94.0 | 97.0 | **86.2** |

Table 5: Results for general DG tasks.

| Domain | PACS | | | | | VLCS | | | | |
|---|---|---|---|---|---|---|---|---|---|---|
| | A | C | P | S | Avg | C | L | S | V | Avg |
| IRM | 85.7 ± 1.0 | 79.3 ± 1.1 | 97.6 ± 0.4 | 75.9 ± 1.0 | 84.6 | 97.6 ± 0.5 | 64.7 ± 1.1 | 69.7 ± 0.5 | 76.6 ± 0.7 | 77.2 |
| Group-DRO | 88.2 ± 0.7 | 82.4 ± 0.8 | 97.7 ± 0.2 | 80.6 ± 0.9 | 87.2 | 97.8 ± 0.0 | 66.4 ± 0.5 | 68.7 ± 1.2 | 76.8 ± 1.0 | 77.4 |
| MIXUP | 87.4 ± 1.0 | 80.7 ± 1.0 | 97.9 ± 0.2 | 79.7 ± 1.0 | 86.4 | 98.3 ± 0.3 | 66.7 ± 0.5 | 73.3 ± 1.1 | 76.3 ± 0.8 | 78.7 |
| DANN | 86.4 ± 1.4 | 80.6 ± 1.0 | 97.7 ± 0.2 | 77.1 ± 1.3 | 85.5 | 95.3 ± 1.8 | 61.3 ± 1.8 | 74.3 ± 1.0 | 79.7 ± 0.9 | 77.7 |
| Unit-DRO | **88.3 ± 0.1** | **84.8 ± 0.1** | 96.4 ± 0.1 | **82.1 ± 0.1** | **87.9** | **98.1 ± 0.1** | **68.0 ± 0.0** | 71.3 ± 0.1 | **78.8 ± 0.0** | **79.1** |

**Other DG-ReID Protocols.** In addition to the most popular evaluation protocols used in Gulrajani & Lopez-Paz (2020) and Tab. 12, we also compare Unit-DRO with other methods using the following protocols: (1) one-to-multiple setting Jin et al. (2020); (2) multiple-to-one setting Dai et al. (2021b); (3) multiple-to-one setting Zhao et al. (2021); and (4) multiple-to-multiple setting Jin et al. (2020). We summarize the difference between different protocols in Tab. 2. As shown in Tab. 11, Unit-DRO outperforms other methods with a clear margin in both average mAP and Rank-1 accuracy. Due to the page limit, please also see the results of protocols (2)∼(4) in Appendix, which demonstrate the robustness of the proposed Unit-DRO across different evaluation protocols. Besides, due to privacy issues, the Duke dataset is not appropriate for use. We thus conduct experiments under different evaluation protocols but remove the Duke from source/target domains, and Table 4 shows that the performance margin between Unit-DRO and other baselines becomes larger. Because these protocols are used in different DG-ReID papers, we choose the SOTA method under every protocol for comparison. The privacy issue is also the motivation of our DGWD setting, where images are randomly sampled and are harder to identify which domain it is from.

**General Domain Generalization.** Apart from person ReID, we also compare Unit-DRO with other general domain generalization methods, including IRM Arjovsky et al. (2020), Group-DRO Sagawa et al. (2019), DANN Ganin et al. (2016), and Mixup Yan et al. (2020). For a fair comparison, we use test-domain validation, which is one of the most important methods in Gulrajani & Lopez-Paz (2020). Specifically, this strategy is an oracle selection one since we choose the model maximizing the accuracy on a validation set that follows the distribution of the test domain. As shown in Table. 5, Unit-DRO consistently outperforms all baseline methods for general domain generalization tasks with a clear margin without using demographics.

Table 4: Comparison with SOTA DG-ReID methods under different evaluation protocols, where the Duke is removed from source and target domains. The best accuracy is highlighted by **bold**.

| Protocol (i) | mAP | Rank-1 | Protocol (ii) | mAP | Rank-1 |
|---|---|---|---|---|---|
| SNR | 54.3 | 48.48 | RaMoE | 31.2 | 32.4 |
| Ours | **58.84** | **52.7** | Ours | **41.7** | **40.4** |

| Protocol (iii) | mAP | Rank-1 | Protocol (iv) | mAP | Rank-1 |
|---|---|---|---|---|---|
| M3L | 26.7 | 27.9 | RaMoE | 68.8 | 58.9 |
| Ours | **27.8** | **29.1** | Ours | **73.2** | **65.4** |

### 4.3 ANALYSIS

**Ablation studies. Choosing the best $|\mathcal{M}|$.** $|\mathcal{M}|$ denotes the size of memory. $|\mathcal{M}| = 0$ means that we use the batch data as the normalizer $\mathbb{E}_{\mathcal{P}}[e^{\ell(x,y;\theta)/\tau^*}]$, which cannot attain the best performance (line 3,5 in the table). A huge $\mathcal{M}$ contains much out-of-date data and also attains inferior results (line 7). In our experiments, we propose to use $|\mathcal{M}| = 10 \times bsz$, where $bsz$ is the batch size during training.

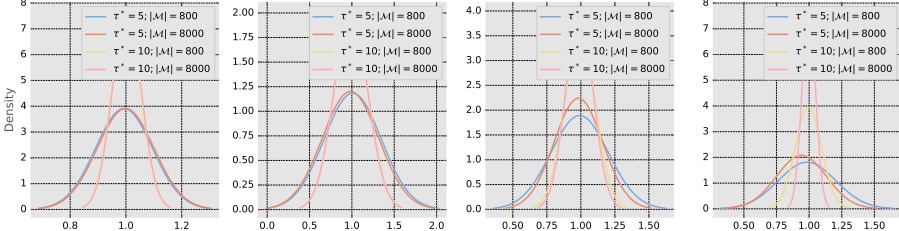

Figure 4: Visualizing the distribution of the sample weight at $1k, 5k, 10k, 20k$ steps, respectively (from left to right). The horizontal axis is the value of the weight and the vertical axis is the density.

**Choosing the best $\tau^*$.** Actually, the selection of multi-step $\tau^*$ is not complicated. At the first few epochs, we simply set $\tau^*$ to a large value such as $100$. In epochs 40, and 70, we decay it to smaller values. When $|\mathcal{M}| = 800$, the performance gap will not be much sensitive to different choices of $\tau^*$. **Sensitivity**. As shown in line 3, even with a constant $\tau^*$ and without a memory bank, the performance of Unit-DRO beats both ERM and KL-DRO by a large margin, which is not too sensitive. To avoid grid search for hyper-parameters, we further propose an alternative approach (**Linear-decay $\tau^*$**), which models $\tau^*$ as a decreasing function of training steps. Specifically, $\tau^* = 100(1 - \frac{t}{T+1})$, such a method attains 65.0 R-1 accuracy and 72.3 mAP, which is comparable to the grid search result and simpler.

**Sample Weights**. Considering that the proposed Unit-DRO will upweight and downweight different samples, we thus visualize the distribution of sample weight to better understand the influences of different components. Specifically, during training, we save the mean and variance of sample weights for every 1k iterations/steps. We assume these weights follow the Gaussian distribution $\mathcal{N}(\mu, \delta)$ and plot diagrams based on the mean $\mu$ and variance $\delta$. The $x$-coordinate of these diagrams is just the value between $[\mu - 3 * \delta, \mu + 3 * \delta]$, not the real values of weights. Based on the loss values of each sample, we calculate the weights under the following two settings: 1) **sample weights without the weight queue.** In this case, these weights are normal-

Table 6: Ablation studies on different Unit-DRO components.

| Line | $\tau^*$ | $|\mathcal{M}|$ | R-1 | mAP |
|------|----------|-----------------|------|------|
| 1 | KL-DRO | KL-DRO | 35.7 | 41.2 |
| 2 | ERM | ERM | 57.8 | 66.2 |
| 3 | 20 | 0 | 63.6 | 71.5 |
| 4 | [50,5,3] | 800 | 64.2 | 72.1 |
| 5 | [100, 5, 3] | 0 | 63.4 | 71.6 |
| 6 | [100, 5, 3] | 800 | 65 | 72.2 |
| 7 | [100, 5, 3] | 4000 | 63.9 | 71.9 |
| **8** | **[100,20,3]** | **1600** | **64.2** | **72** |

ized in their batches, so the mean of all distributions here is 1. As shown in Figure 3, we have already discussed this setting in the former sections; 2) **sample weights with different length of weight queue $|\mathcal{M}|$.** In Figure 4, we show the distribution of sample weight at $1k, 5k, 10k, 20k$ training steps to indicate how the weight distribution changes during training. Intuitively, we need a large $|\mathcal{M}|$ to better estimate $\mathbb{E}_\mathcal{P}[e^{\ell(x,y;\theta)/\tau^*}]$. However, as $|\mathcal{M}|$ becomes larger, the estimation becomes inaccurate. For example, we consider an extreme case: $|\mathcal{M}| = T - 1$, then the queue absolutely contains all training data. Therefore, it is catastrophic to estimate $\mathbb{E}_\mathcal{P}[e^{\ell(x,y;\theta)/\tau^*}]$ in step $T$ by such a queue. The large queue contains too many old weights which are unsuitable for the current model. Figure 4 depicts the phenomenon, where the distribution with a larger $|\mathcal{M}|$ has smaller $\mu$. See Appendix E.2 for more visualization results and discussions about the distribution diagrams of the multi-step $\tau^*$.

**Visualization using $t$-SNE.** We compare the proposed Unit-DRO with MetaBIN and DualNorm through $t$-SNE visualization. We observe a distinct division of different domains in Figure 5a, which indicates that a domain-specific feature space is learned by the DualNorm. MetaBIN and the proposed Unit-DRO tackle this problem well and the overlaps in Figure 5b and Figure 5c between different domains are more prominent. With $t$-SNE visualization, we see that Unit-DRO can learn domain-invariant representations while keeping discriminative capability for ReID tasks. However, MetaBIN follows a meta-learning pipeline and requires extra expensive demographics. In contrast, the proposed framework Unit-DRO is much simpler without using any demographics. We also provide more visualization results and analysis of the discriminative capability in Section E.3 of Appendix.

**Domain Divergence.** We explore MMD distance Tolstikhin et al. (2016) or $\mathcal{A}$-distance Long et al. (2015) as the measure of domain discrepancy Ben-David et al. (2010). As shown in Table 7, we find that Unit-DRO can learn comparable or even more invariant representations compared to MetaBIN, which outperforms DualNorm by a large margin. We also study the correlation between the weights

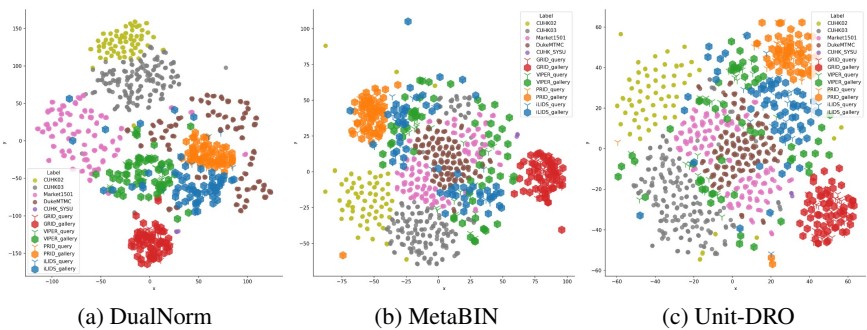

|           |           |           |
|:---------:|:---------:|:---------:|
| (a) DualNorm | (b) MetaBIN | (c) Unit-DRO |

Figure 5: Visualization of the embeddings on training and test datasets. Query and gallery samples of these unseen datasets are shown using different types of mark. Best viewed in color.

Table 7: Divergence measurement on unseen datasets (U), training datasets (T), and all datasets (A).

| Method | MMD↓ (U) | MMD↓ (T) | MMD↓ (A) | $\mathcal{A}$ ↓ (U) | $\mathcal{A}$ ↓ (T) | $\mathcal{A}$ ↓ (A) |
|---|---|---|---|---|---|---|
| DualNorm | 0.52 | 0.21 | 0.41 | 1.96 | 1.91 | 1.88 |
| MetaBIN | 0.41 | 0.19 | 0.36 | 1.96 | 1.89 | 1.86 |
| Unit-DRO | 0.41 | 0.19 | 0.35 | 1.95 | 1.89 | 1.85 |

for each dataset and the MMD distance. For each dataset, we calculate the sum of MMD distance between it to all other datasets. Besides, we calculate the average weights of the final model for each dataset. Table 8 shows that for a tough dataset (*e.g.,* , CUHK02) that has a large divergence to other datasets, Unit-DRO assigns a relatively higher average weight. *This phenomenon depicts that even without demographics, Unit-DRO can also find meaningful subgroups and upweight them.* We can also see that Unit-DRO upweights samples in CUHK-SYSU which has a relatively small MMD distance with other datasets. It is because the generalization ability is not only dependent on domain divergence, but also on some other factors. We discuss these influence factors and perform error set analysis in Section E.6 of Appendix. We also plot the MMD distance for every pair of datasets and give further analysis in Section E.5 of Appendix.

## 5 CONCLUSION AND FUTURE WORK

Traditional DG-ReID methods fail to work in the cases where domain information, such as camera labels, or other demographics, are not available due to security and privacy issues. To this end, we introduce DGWD-ReID, a more general setting that requires the model to learn domain-invariant representations without demographics. To address this problem, we propose Unit-DRO, which is a simple yet effective algorithm that substantially improves the model generalization performance without requiring expensive demographics during training. Extensive experimental results demonstrate that the proposed Unit-DRO not only achieves comparable or better performance compared with other DG-ReID methods using the demographic but also attains superior generalization capability on general domain generalization applications.

Different from the typical classification datasets, where domains are usually partitioned by image styles, ReID datasets have more fine-grained variation factors, *e.g.,* the styles of images, camera perspective changes within one dataset, and the shooting conditions at different times on the same camera. We believe that simply specifying each dataset as a domain is suboptimal and better domain inference methods that consider the above variation factors will be the subject of future study.

Table 8: Average weight and one-to-all MMD distance for training datasets.

|        | Cuhk02 | Cuhk03 | Market | Duke | SYSU |
|--------|--------|--------|--------|------|------|
| Weight | 1.02   | 0.99   | 0.99   | 1.00 | 1.01 |
| MMD    | 1.66   | 1.17   | 1.15   | 1.16 | 1.04 |

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

# Generalizable Person Re-identification Without Demographics
## – Appendix –

## A  SUPPLEMENTARY DISADVANTAGES OF DG REID SETTING

1. **Privacy Risks on using demographics in ReID tasks**. ReID is a research about person which is naturally with higher requirements for privacy. Utilizing demographic information will bring the following two concerns. 1) Although the demographic data used in current ReID research is simple, e.g., the different campuses in Market1501 and CUHK, the demographic information in practical ReID deployments is much richer and can be obtained easily based on some inherent physical properties of cameras (such as network cameras' MAC address and geographical locations). Thus, the utilization of finer and richer physical properties of cam-



ViPeR    Market-1501    DukeMTMC

*Different Cams*

Figure 6: Samples on ReID datasets.

eras will increase the leakage risk of the privacy information of pedestrians. 2) The social relationships between different people are also possibly to be exposed by demographic information, which may be more sensitive than geographical information. For example, if two-person IDs's images are marked with the same camera, their social relationships may be easily measured by counting their occurrence frequency in all cameras. In summary, the uses of demographic data in real-world ReID applications have evident risks for personal privacy. Thus, we propose the setting of DGWD- ReID, where only a large-scale gallery of pedestrian images without any demographic data, e.g., camera IDs, can be used for training ReID models. Based on the setting of DGWD- ReID, researchers will be forced to **exploit invariant features from the training data itself, rather than resort to the side information**, i.e., demographic data, so as to reduce the risks of privacy leakage in ReID applications. We also agree that the annotation of subject ID is much more expensive and has more risks to personal privacy. We will tackle the challenging problem of unsupervised DG-ReID in future work. Thanks for your valuable suggestion again!

2. **The importance of DGWD setting in ML.** Learning generalizable models without domain labels is becoming an important matter of concern in the ML community Creager et al. (2021); Kim & Lee (2021). Our algorithm can also be applied to generalized DG tasks where environment labels are unavailable at training time, either because they are difficult to obtain or due to privacy.

3. Finally, a previous study Srivastava et al. (2020) shows that **how to optimally partition domains in training images that can benefit generalization capability most is still unclear**, which indicates the direct use of demographic data as domain labels may be inferior for learning domain-invariant representations. Thus, it is also a promising topic for DGWD-ReID as we discussed in the conclusion. In the fairness community, existing work has found that designing rational methodology can find domains that are maximally informative for downstream invariance learners. These domain IDs, and camera IDs make sense for humans, however, the similarities between different domains/camera views vary greatly. How to find more optimal domain partitions for ReID tasks is still an open problem.

## B  EXTENDED RELATED WORK

**Domain generalization.** Domain/Out-of-distribution generalization Muandet et al. (2013); Zhang et al. (2021a) aims to learn a model that can extrapolate well in unseen environments. Representative methods like Invariant Risk Minimization (IRM) Arjovsky et al. (2020) and its variant Ahuja et al. (2020) are recently proposed to tackle this challenge. IRM centers on the objective of extracting data representations that lead to invariant prediction across environments under a multi-environment setting. The main difference here is that we propose to learn invariant representations without demographics.

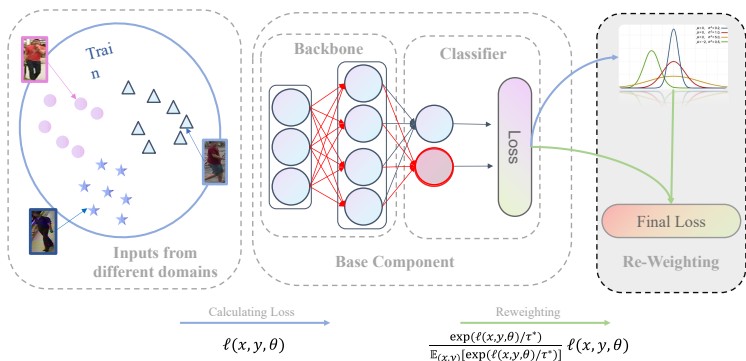

Figure 7: An illustration of DGWD-ReIDthat propose to reweight training instances without demographic information.

**Unsupervised-domain adaptation Person ReID.** Unsupervised Domain Adaptation (UDA) technologies have great progress Peng et al. (2020) and have been widely adopted for cross-domain person ReID. The UDA-based ReID methods usually attempt to transfer the knowledge learned from the labeled source domains to target domains, depending on target-domain images Luo et al. (2020); Huang et al. (2020), features Wang et al. (2018) or metrics Peng et al. (2016). Another group of UDA-based methods Ge et al. (2020); Zhai et al. (2020) propose to explore hard or soft pseudo labels in the unlabeled target domain using its data distribution geometry. Though UDA-based methods improve the performance of cross-domain ReID to a certain extent, most of them require a large amount of unlabeled target data for model retraining.

**Distributionally Robust optimization.** Distributionally Robust optimization Ben-Tal et al. (2009) solve robust versions of ERM, which replace the expected risk under the training data distribution with the worst expected risk over a pre-defined uncertainty set $\mathcal{Q}$ (refer to Rahimian & Mehrotra (2019) for a review). Recent studies constitute $\mathcal{Q}$ analytically, such as using moment constraint Delage & Ye (2010); Nguyen et al. (2020), $f$-divergence Hu & Hong (2013); Michel et al. (2021), Wasserstein/MMD ball Sinha et al. (2017); Staib & Jegelka (2019) or coarse-grained mixture models Oren et al. (2019); Duchi et al. (2019). We reformulate KL-constraint DRO to an important sampling problem (Unit-DRO) and propose an efficient implementation, which scales to large datasets and overparameterized neural networks.

## C  OPTIMIZATION ALGORITHM AND FRAMEWORK DIAGRAM

The optimization algorithm is shown in Algorithm 1 and the diagram is shown in Figure 7.

---

**Algorithm 1:** Online optimization algorithm for Unit-DRO.

**Input:** training data $\mathcal{P}$, batch size $N$, learning rate $\eta$, training iterations $T$.
**Initial**: model parameters $\theta^0$ and weight queue $\mathcal{M}^0 = \{1\}_{i=1}^{M}$.
**for** $t = 1, \dots, T$ **do**
 $(x_i, y_i)_{i=1}^{N} \sim \mathcal{P}$            //Data sampling
 $\mathcal{L} = \frac{1}{N} \sum_{i=1}^{N} \left( \frac{e^{\ell(x,y;\theta^{t-1})/\tau^*(t)}}{\frac{1}{|\mathcal{M}|} \sum_{w_i \in \mathcal{M}} (w_i)} \ell(x, y; \theta^{t-1}) \right)$ //Calculate the reweighted loss
 $\mathcal{M}^t = \left[ \mathcal{M}^{t-1}[N :], \{e^{\ell(x_i, y_i; \theta^{t-1})/\tau^*(t)}\}_{i=1}^{N} \right]$ //Update the weight queue
 $\theta^t \leftarrow \text{SGD} \left( \mathcal{L}, \theta^{t-1}, \eta \right)$            //Update model parameters
**end**

---

# D  DETAILED DATASET SETTING

## D.1  DATASET DETAILS

Details of the training datasets are summarized in Table 9 and the test datasets are summarized in Table 10. All the assets (*i.e.,* datasets and the codes for baselines) we use include an MIT license containing a copyright notice and this permission notice shall be included in all copies or substantial portions of the software.

## D.2  EVALUATION PROTOCOLS

**GRID** Liu et al. (2012) contains 250 probe images and 250 true match images of the probes in the gallery. Besides, there are a total of 775 additional images that do not belong to any of the probes. We randomly take out 125 probe images. The remaining 125 probe images and $1025(775 + 250)$ images in the gallery are used for testing.

**i-LIDS** Wei-Shi et al. (2009) has two versions, images and sequences. The former is used in our experiments. It involves 300 different pedestrian pairs observed across two disjoint camera views 1 and 2 in public open space. We randomly select 60 pedestrian pairs, two images per pair are randomly selected as probe image and gallery image respectively.

**PRID2011** Hirzer et al. (2011) has single-shot and multi-shot versions. We use the former in our experiments. The single-shot version has two camera views $A$ and $B$, which capture 385 and 749 pedestrians respectively. Only 200 pedestrians appear in both views. During the evaluation, 100 randomly identities presented in both views are selected, the remaining 100 identities in view $A$ constitute the probe set, and the remaining 649 identities in view $B$ constitute the gallery set.

**VIPeR** Gray et al. (2007) contains 632 pedestrian image pairs. Each pair contains two images of the same individual seen from different camera views 1 and 2. Each image pair was taken from an arbitrary viewpoint under varying illumination conditions. To compare to other methods, we randomly select half of these identities from camera view 1 as probe images and their matched images in view 2 as gallery images.

We follow the single-shot setting. The average rank-k (R-k) accuracy and mean Average Precision ($m$AP) over 10 random splits are reported based on the evaluation protocol

For the general DG setting, we use two datasets

**PACS** Li et al. (2017) includes $9,991$ images with 7 classes $y \in \{$ dog, elephant, giraffe, guitar, horse, house, person $\}$ from 4 domains $d \in \{$art, cartoons, photos, sketches$\}$.

**VLCS** Torralba & Efros (2011) is composed of 10,729 images, 5 classes $y \in \{$ bird, car, chair, dog, person $\}$ from domains $d \in \{$Caltech101, LabelMe, SUN09, VOC2007$\}$.

## D.3  BASELINES

The baselines in Table. 11 follow the settings in Jin et al. (2020), where

1. A-IN: a naive model where we replace all the Batch Normalization(BN) layers in Baseline by Instance Normalization(IN).

2. IBN: we add IN only to the last layers of Conv1 and Conv2 blocks of Baseline respectively.

3. A-SN: a model where we replace all the BN layers in the Baseline by Switchable Normalization (SN). SN Luo et al. (2019b) can be regarded as an adaptive ensemble version of normalization techniques of IN, BN, and LN (Layer Normalization).

4. IN: four IN layers are added after the first four convolutional blocks/stages of Baseline respectively.

Table 9: Training Datasets Statistics.

| Dataset | IDs | Images |
|---|---|---|
| CUHK02 | 1,816 | 7,264 |
| CUHK03 | 1,467 | 14,097 |
| DukeMTMC-Re-Id | 1,812 | 36,411 |
| Market-1501 | 1,501 | 29,419 |
| CUHK-SYSU | 11,934 | 34,547 |

Table 10: Testing Datasets statistics.

| Dataset | Probe | | Gallery | |
|---|---|---|---|---|
| | Pr. IDs | Pr. Imgs | Ga. IDs | Ga. imgs |
| PRID | 100 | 100 | 649 | 649 |
| GRID | 125 | 125 | 1025 | 1,025 |
| VIPeR | 316 | 316 | 316 | 316 |
| i-LIDS | 60 | 60 | 60 | 60 |

# E    EXTEND EXPERIMENTAL RESULTS

## E.1    CROSS-DOMAIN REID PERFORMANCE

For cross-domain evaluation, we use the Market1501 dataset and DukeMTMC-ReID as the source/target domains iteratively. For example, "Market-Duke" indicates that the labeled source domain is Market1501 and DukeMTMC-ReID is the unseen target domain. Since the style variation within a single dataset is relatively small, previous DG-ReID methods must utilize fine-grained demographics, *e.g.,* camera labels Zhang et al. (2021b), or carefully tune all hyperparameters Choi et al. (2021). Similar to DG-ReID setting, here Unit-DRO does not use any demographic information. As shown in Table 12, we see that the proposed Unit-DRO even achieves a slightly better performance than recent state-of-the-art CD-ReID methods using demographics, suggesting the good cross-domain generalizability of Unit-DRO.

## E.2    DISTRIBUTION DIAGRAMS OF STEP $\tau^*$

Compared to a constant $\tau^*$, weights with step $\tau^*$ always have low $\delta$ and are more stable.

## E.3    ADDITIONAL $t$-SNE VISUALIZATION RESULTS

Figure 9 shows the $t$-SNE results on four unseen datasets. Figure 10 shows the $t$-SNE results on five training datasets and Figure 12 shows the $t$-SNE results on the Market-Duke benchmark. All of these results demonstrate a common pattern, DualNorm Jia et al. (2019) retains large domain divergences and its embedding vector is far from "domain invariant". MetaBIN Choi et al. (2021) utilizes a complex framework and expensive demographics, which is able to reduce domain divergences. Unit-DRO achieves a comparable or even better result than MetaBIN Choi et al. (2021) in a simpler and cheaper paradigm. **Consider discriminative capability.** Figure 11 visualizes the probe and gallery samples on four test datasets individually. The utopian discrimination result is that every query-galley pair has the closest intra-identity distance and a relatively large inter-identity distance. Figure 11d and Figure 11b shows that Unit-DRO performs well matching on the i-LIDS and the PRID dataset. However, we observe an interesting phenomenon, termed "Inter-Identity Cluster". Specifically, probes and galleries of different identities came together in some clusters. These clusters are always seen on the VIPeR and the GRID datasets (Figure 11a and Figure 11b), which reveals why Unit-DRO performs much poorly on these two datasets.

## E.4    IMPLEMENTATION OF DOMAIN DIVERGENCE MEASUREMENT

In general, MMD distance Tolstikhin et al. (2016) is defined by the idea of representing distances between distributions as distances between mean embeddings of features. Following MMFA model Lin

Table 11: Comparison with recent DG-ReID methods using the protocol (1).

| Source | Methods | Avg | | Target:Market1501 | | Target:Duke | | Target:PRID | | Target:GRID | | Target:VIPeR | | Target:iLIDs | |
|---|---|---|---|---|---|---|---|---|---|---|---|---|---|---|---|
| | | mAP | Rank-1 | mAP | Rank-1 | mAP | Rank-1 | mAP | Rank-1 | mAP | Rank-1 | mAP | Rank-1 | mAP | Rank-1 |
| Market1501 | A-IN | 45.2 | 44.1 | 75.3 | 89.8 | 24.1 | 42.7 | 33.9 | 21 | 35.6 | 27.2 | 38.1 | 29.1 | 64.2 | 55 |
| | IBN | 39.9 | 39.1 | 81.1 | 92.2 | 21.5 | 39.2 | 19.1 | 12 | 27.5 | 19.2 | 32.1 | 23.4 | 58.3 | 48.3 |
| | A-SN | 42.2 | 40.9 | 83.2 | 93.9 | 20.1 | 38 | 35.4 | 25 | 29 | 22 | 32.2 | 23.4 | 53.4 | 43.3 |
| | IN | 45.7 | 45.1 | 79.5 | 90.9 | 25.1 | 44.9 | 35 | 25 | 35.7 | 27.8 | 35.1 | 27.5 | 64 | 54.2 |
| | SNR | 50.9 | 49.6 | **84.7** | **94.4** | 33.6 | 55.1 | 42.2 | 30 | 36.7 | 29 | 42.3 | 32.3 | 65.6 | 56.7 |
| | Ours | 54.7 | 53.2 | 83.5 | 92.2 | 33.8 | 55.5 | 56.7 | 44.5 | 40 | 31 | 44.7 | 35.3 | 69.3 | 60.7 |
| Duke-MTMC | A-IN | 41.2 | 43.6 | 21.8 | 56 | 64.5 | 78.9 | 38.6 | 29 | 19.6 | 13.6 | 35.1 | 27.2 | 67.4 | 56.7 |
| | IBN | 39.9 | 41.7 | 26.5 | 52.5 | 69.5 | 81.4 | 27.4 | 19 | 19.9 | 12 | 32.8 | 23.4 | 63.5 | 61.7 |
| | A-SN | 42.3 | 45.5 | 24.6 | 55 | **73** | **85.9** | 41.4 | 32 | 18.8 | 12.8 | 31.3 | 24.1 | 64.8 | 63.3 |
| | IN | 43.7 | 45.1 | 27.2 | 58.5 | 68.9 | 80.4 | 40.5 | 27 | 20.3 | 13.2 | 34.6 | 26.3 | 70.6 | 65 |
| | SNR | 51.3 | 52.2 | 33.9 | 66.7 | 72.9 | 84.4 | 45.4 | 35 | 35.3 | 26 | 41.2 | 32.6 | **79.3** | **68.7** |
| | Ours | 55.6 | 56.2 | 36.4 | 69.2 | 72.8 | 81.7 | 63.2 | 53.23 | 39.9 | 30.4 | 44.5 | 34.8 | 76.7 | 68 |

Table 12: Comparison with recent state-of-the-art CD-ReID methods.

| Method | Market-Duke | | | | Duke-Market | | | |
|---|---|---|---|---|---|---|---|---|
| | R-1 | R-5 | R-10 | mAP | R-1 | R-5 | R-10 | mAP |
| CrossGrad | 48.5 | 63.5 | 69.5 | 27.1 | 56.7 | 73.5 | 79.5 | 26.3 |
| QAConv | 48.8 | - | - | 28.7 | 58.6 | - | - | 27.6 |
| L2A-OT | 50.1 | 64.5 | 70.1 | 29.2 | 63.8 | 80.2 | 84.6 | 30.2 |
| OSNet-AIN | 52.4 | 66.1 | 71.2 | 30.5 | 61.0 | 77.0 | 82.5 | 30.6 |
| SNR | 55.1 | - | - | 33.6 | 66.7 | - | - | 33.9 |
| DIR-ReID | 54.5 | 66.8 | 72.5 | 33.0 | 68.2 | 80.7 | 86.0 | 35.2 |
| MetaBIN | 55.2 | 69.0 | 74.4 | 33.1 | 69.2 | 83.1 | 87.8 | 35.9 |
| **Unit-DRO** | **55.5** | **70.3** | **74.9** | **33.8** | **69.2** | **83.7** | **88.0** | **36.4** |

et al. (2018), we use the RBF characteristic kernel with bandwidth $\alpha_2 = 1 : 5 : 10$ to compute the MMD distance. $\mathcal{A}$-distance Long et al. (2015) can be approximated as $d_{\mathcal{A}}(d_i, d_j) = 2(1 - 2\sigma)$, where $\sigma$ is the error of a two-sample classifier distinguishing features of samples from two distinct domains $d_i, d_j$. Note that we have not only two domains. To measure the $\mathcal{A}$-distance or MMD-distance on four unseen datasets, we calculate the average mean distance of each domain pair, namely

$$\mathcal{A}(U) = \frac{1}{6} \sum_{i=1}^{4} \sum_{j=i+1}^{4} \mathcal{A}(d_i, d_j). \tag{9}$$

## E.5 Additional domain divergence measurement results

The MMD distance between every dataset pair of all the datasets is plotted in Figure 13a. The MMD distance between every dataset pair of five training datasets is shown in Figure 13b and that of four test datasets is shown in Figure 13c. For the training dataset, we find that the CUHK02 dataset remains large divergences with almost all the other domains. Namely, the CUHK02 dataset is more likely to be an out-of-distribution dataset and is more important to generalization capability. Hence, Unit-DRO assigns relatively higher weights for samples in the CUHK02 dataset. In terms of test datasets, the GRID dataset maintains the largest MMD distance among these datasets. It is also the reason why Unit-DRO performs badly on the GRID dataset. However, domain divergence is not the only factor that affects generalization performance. Figure 13c shows that the PRID dataset has a larger domain divergence than VIPeR. However, Unit-DRO performs better on the PRID dataset than on the VIPeR dataset. We exploit the underlying reasons in Section E.6.

## E.6 Error set analysis

We select some successfully retrieved pairs[2] and failure cases in Figure 14. We plot query images and corresponding gallery images at the top and bottom of these figures respectively. Figure 14a

---

[2]We name a query-gallery images pair "successfully retrieved pair" such that the distance between the query image and its corresponding gallery image is the closest in all of the gallery images. Other pairs are named failure cases.

Table 13: Comparisons against state-of-the-art DG methods for person ReID on evaluation protocol (ii) and (iii). Protocol (ii) and (iii) are both multiple-to-one setting which used in RaMoE Dai et al. (2021b) and M3L Zhao et al. (2021) respectively. Unit DRO beats them in both these two settings.

| | Protocol (ii) | | | | Protocol (iii) | | |
|---|---|---|---|---|---|---|---|
| Method | mAP | Rank-1 | Rank-5 | Rank-10 | Method | mAP | Rank-1 |
| RaMoE | 35.5 | 36.6 | 54.3 | 64.6 | M3L | 29.9 | 30.7 |
| Ours | **43.8** | **43.6** | **65.3** | **74.5** | Ours | **30.9** | **31.1** |

Table 14: Comparisons against state-of-the-art DG methods for person ReID on evaluation protocol (iv). Unit DRO outperforms RaMoE Dai et al. (2021b) in protocols (iv) by a large margin.

| Method | Avg | | Target:PRID | | Target:GRID | | Target:VIPeR | | Target:iLIDs | |
|---|---|---|---|---|---|---|---|---|---|---|
| | mAP | Rank-1 | mAP | Rank-1 | mAP | Rank-1 | mAP | Rank-1 | mAP | Rank-1 |
| SNR | 64.6 | 55.4 | 60.0 | 49.0 | 41.3 | 30.4 | 65.0 | 55.1 | 91.9 | 87.0 |
| RaMoE | 71.3 | 63.0 | 66.8 | 56.9 | 53.9 | 43.4 | 72.2 | 63.4 | 92.3 | 88.4 |
| Ours | **76.1** | **68.0** | **79.4** | **71.3** | **59.8** | **50.2** | **77.1** | **68.9** | 88.2 | 81.7 |

shows that query and gallery images in the failure case have a relatively large view change (front and back shooting). In contrast, successfully retrieved query-gallery pairs in Figure 14b have almost the same camera view. This result shows that Unit-DROcannot well overcome the challenges brought by changes in the camera view. Namely, we can leverage advanced structure in supervised ReID methods to eliminate the sensitivity of Unit-DRO to camera perspective. Figure 14c and Figure 14b show that the camera perspective changes between query and gallery set in the PRID dataset are small, which is one of the reasons why Unit-DRO performs much better on the PRID dataset than the GRID dataset[3]. According to error set analysis, we can explain the phenomenon mentioned in Section E.5 that Unit-DRO performs superior on the dataset with a relatively high domain divergence (the PRID dataset) than the dataset with low domain divergence (the VIPeR dataset). Figure 14e shows that query-gallery pairs in the VIPeR dataset always maintain camera view changes that more than $90°$, which is harder to identify compared to the PRID dataset. Finally, the i-LIDS dataset has the lowest MMD distances among other datasets and the camera perspective changes between its query-gallery pairs are always small. These good properties enable Unit-DRO to achieve a rank-1 accuracy of 80.7 on the i-LIDS dataset. So far, we can conclude that all of the domain style divergence, intrinsic characteristics of datasets (camera perspective changes), and model capacity [4] affect the performance of DG ReID and DGWD-ReID methods.

---

[3]Another reason is the domain divergence, as we discussed in Section E.5.

[4]larger backbones and advanced learning paradigm always attains better generalization capability.

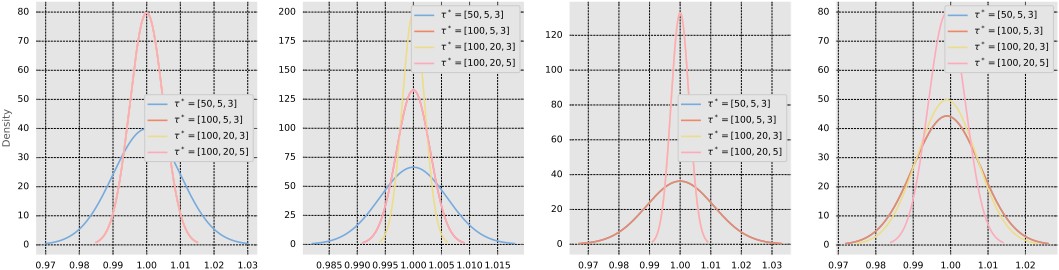

Figure 8: Distribution visualization of sample weights ($|\mathcal{M}| = 800$ by default) of steps $[1000, 50000, 100000, 150000]$ (from left to right). The horizontal axis represents the weight, and the vertical axis represents the density. $\tau^* = [\tau_1, \tau_2, \tau_3]$ means $\tau^* = \tau_1$ initially and decayed to $\tau_2$ and $\tau_3$ at 40 and 70 epochs.

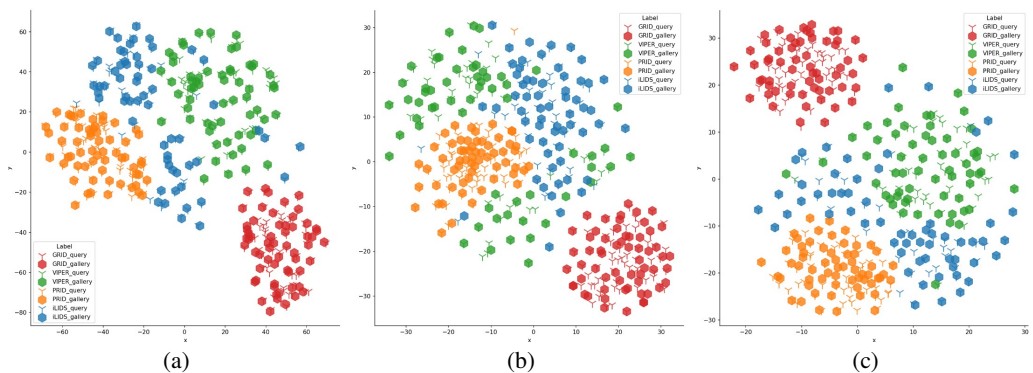

Figure 9: The $t$-SNE visualization of embedding vectors on four unseen target datasets. Query and gallery samples are expressed in different shapes. Best viewed in color.

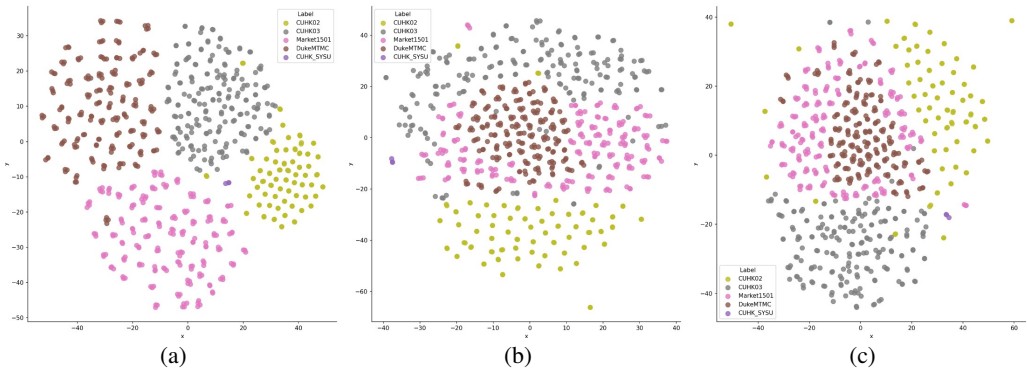

Figure 10: The $t$-SNE visualization of embedding vectors on five training datasets. Best viewed in color.

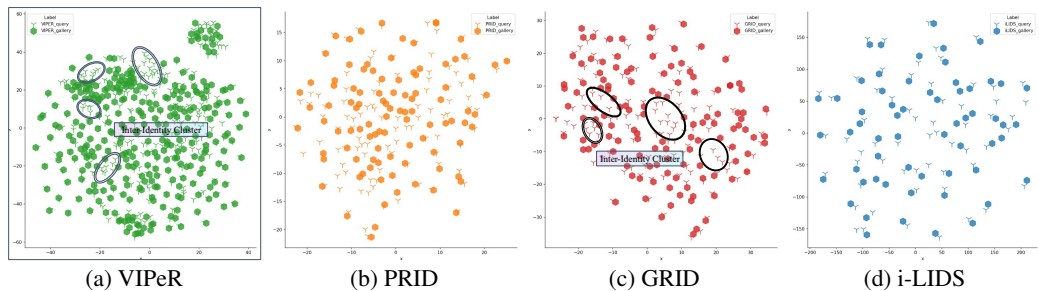

Figure 11: The $t$-SNE visualization of embedding vectors on four test datasets individually. Best viewed in color.

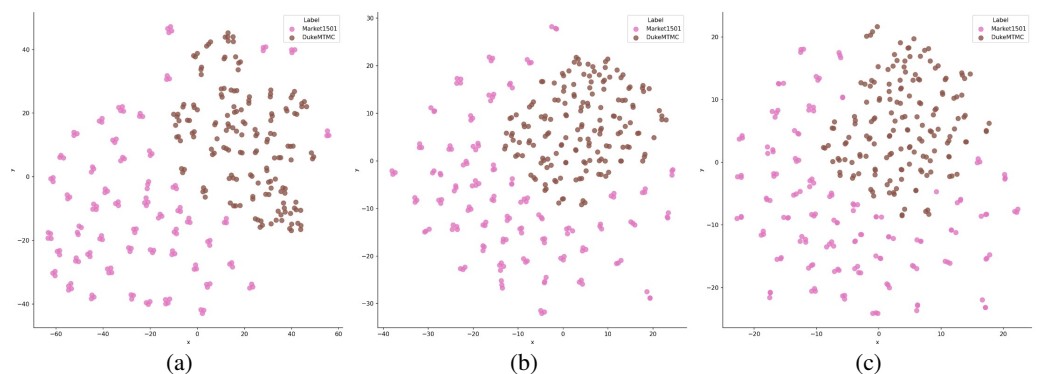

Figure 12: The $t$-SNE visualization of embedding vectors on Market1501 Zheng et al. (2015) and DukeMTMC-ReID Zheng et al. (2017). Models are trained on the Market-Duke benchmark. Best viewed in color.

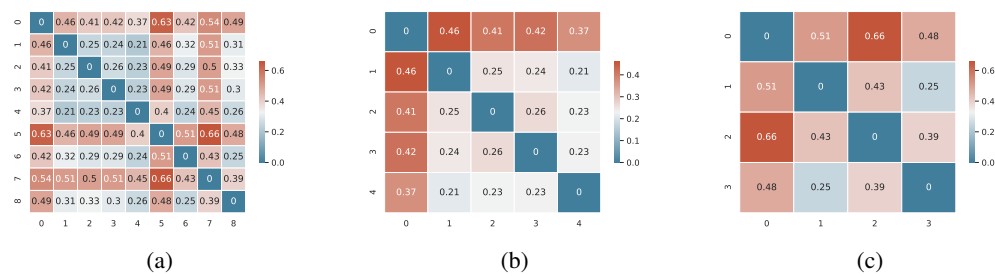

Figure 13: The heatmaps of MMD distance on training and test dataset pairs. (a, b): 0: CUHK02, 1: CUHK03, 2: Market1501, 3: DukeMTMC, 4: CUHK-SYSU, 5: GRID, 6: VIPeR, 7: PRID, 8: i-LIDS. (c): 0: GRID, 1: VIPeR, 2: PRID, 3: i-LIDS.

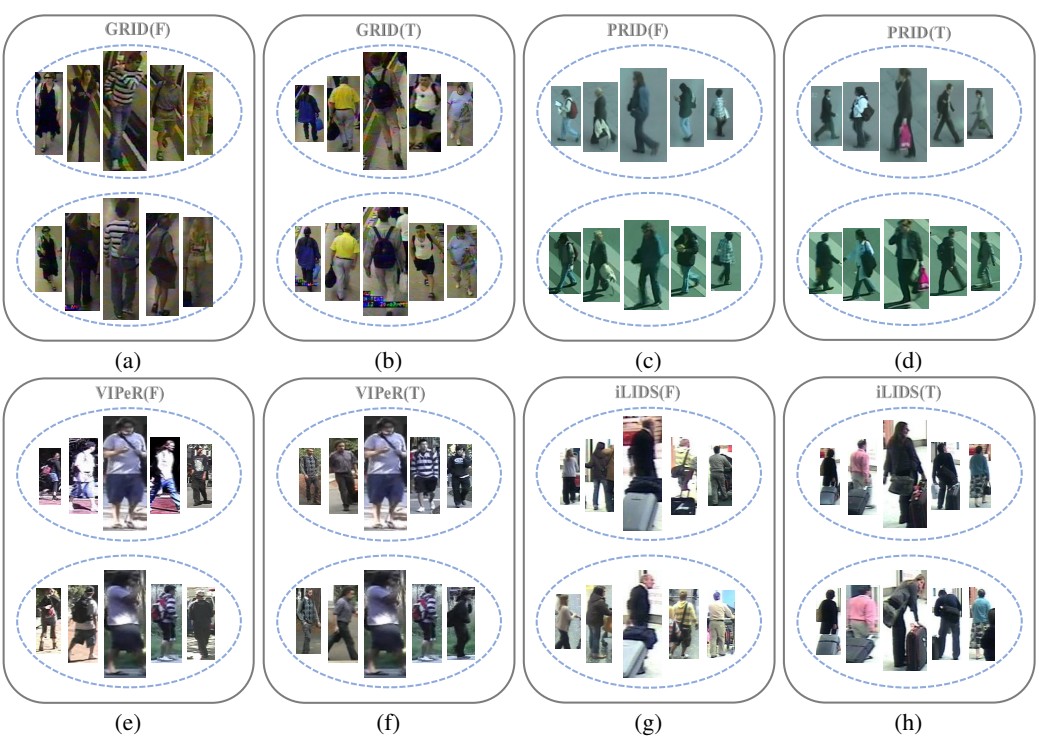

Figure 14: Error set analysis. (a): Failure cases in the GRID datasets. (b) Successfully retrieved pairs in the GRID datasets. (c) Failure cases in the PRID datasets. (d) Successfully retrieved pairs in the PRID datasets. (e): Failure cases in the VIPeR datasets. (f) Successful retrieved pairs in the VIPeR datasets. (g) Failure cases in the i-LIDS datasets. (h) Successfully retrieved pairs in the i-LIDS datasets.

