# OpenReview forum: "Generalizable Person Re-identification Without Demographics"
_ICLR.cc/2023/Conference — Submitted to ICLR 2023_

### Official Review · Reviewer_42Bp · 2022-10-20

**Confidence:** 3
**Correctness:** 3
**Technical Novelty And Significance:** 3
**Empirical Novelty And Significance:** 3
**Recommendation:** 6

**Clarity, Quality, Novelty And Reproducibility:**

Clarity: In general, the authors explain their methods and experiments very clearly, except for weakness(3);

Quality: The quality of writing, experimenting settings, figures, and tables are well;

Novelty: The paper pioneered to address the DGWD-ReID problem and innovates the DRO;

Reproducibility: The model is simple but efficient, and the reproducibility may be well. There is no link to the code in the paper.


**Strength And Weaknesses:**

Strengths:
(1) Demographics information is not used, and the problem studied is more realistic;
(2) The author innovatively puts forward the Unit-DRO and it is simple and efficient;
(3) The author makes an in-depth analysis of the shortcomings of the Unit-DRO and proposes a multi-step solution and a weighted queue to get better performance;
(4) The experimental results are better than other CD-ReID and DG-ReID methods in the absence of demographics.
(5) Mathematical analysis and ablation studies are sufficient.

Weaknesses:
(1) There is a lack of a framework diagram that can visually demonstrate the method proposed;
(2) Partial experimental results are quite worse than the state-of-the-art, like “Source Market1501 & Target Market 1501”;
(3) Some of the figures or tables are not clearly illustrated, like the horizontal axis of each chart in Figure4;
(4) Lack of detailed analysis of the ReID task itself and the adaptability of the proposed method to the ReID task.


**Summary Of The Paper:**

The paper considers domain generalization person re-identification without demographics (DGWD-ReID), which is more difficult than cross-domain person ReID (CD-ReID) and DG-ReID but closer to the real-world scenario because the domain and camera labels are not always available for privacy problem. Innovatively, in order to deal with the uncertainty of domain distribution, the author improved distributionally robust optimization(DRO) by reformulating the minimax optimization as an important sampling problem and named it Unit-DRO. Moreover, the author introduced a multi-step solution and a weighted queue to further improve its performance. The multi-step solution can determine a hyper-parameter adaptively by the training step and the weighted queue can help to estimate the weights better and solve the problem that “stochastic mini-batch training” causes. Sufficient experiments demonstrate that the approach is even better than most approaches with demographics.

**Summary Of The Review:**

The approach is innovative for proposing a simple but efficient method for DG-ReID without demographics, and the mathematical derivation, experimental verification, and ablation studies are sufficient and persuasive. These strengths can convince me to accept the paper. However, the paper still needs some improvement in some respects. Firstly, the author should analyze in more detail the adaptability of its approach to the ReID task. Secondly, the author would better present their method more visually with an overall framework. What’s more, the author should analyze the possible reasons for some poor experimental results.

---

> ### Author Response · Authors · 2022-11-09
> **Response to Reviewer 42Bp**
>
> Thank you for the positive comments and careful reading, you have almost concluded the contributions of our work. We will answer the four concerns point by point.
>
> ----
>
>
> **[Concern 1]** *There is a lack of a framework diagram that can visually demonstrate the method proposed.*
>
> Thanks for your valuable suggestion! We have added a diagram to describe the proposed method in Appendix C which will be updated in the final submission.
>
> **[Concern 2]** *Partial experimental results are quite worse than the state-of-the-art, like “Source Market1501 & Target Market 1501”;*
>
> A native ERM algorithm will achieve the best in this setting because the test distribution is the same as the training distribution. However, our algorithm seeks better generalization ability and there is always a tradeoff between source domain accuracy and unseen target domain accuracy. To verify this point, we also evaluate MetaBIN at this setting and it attains $81.7%$ mAP, which is inferior to our $83.5%$.
>
> **[Concern 3]** *Some of the figures or tables are not clearly illustrated, like the horizontal axis of each chart in Figure4;*
>
> Thanks for pointing out the confusing content. Because Figure 4 is the visualization of distributions of sample weights. Hence the horizontal axis is the value of the sample weight and the vertical axis is the density. We have added a corresponding description,
>
> **[Concern 4]** *Lack of detailed analysis of the ReID task itself and the adaptability of the proposed method to the ReID task.*
>
> We apologize that we cannot totally understand the question, however, we conduct extensive analysis and comparison in this work, including
>
> 1. **Generalization performance on DG-ReID (Domain generalization) and CD-ReID (Cross-Domain) benchmarks** on five evaluation protocols, which shows Unit-DRO attains superior performance on both DG and CD ReID settings.
>
> 2. **t-sne visualization & domain divergence measurement** of the learned representation distribution (Figure .5, Table.8,9), which shows the Unit-DRO can learn domain-invariant representations while keeping discriminative capability for ReID tasks without demographics. Besides, even without demographics, Unit-DRO can also find meaningful subgroups and upweights them.
>
> 3. For the **failure case analysis for ReID datasets**, We analyze the difficulties for ReID tasks for generalization in section D.6. We show that the changes in the camera view and large domain divergence still remain a challenging problem for DG-ReID and we need a more powerful structure or effective algorithms.

---

> > ### Author Response · Authors · 2022-11-14
> > **A Gentle Reminder of Feedbacks**
> >
> > Dear Reviewer 42Bp,
> >
> > Thanks again for your careful reading and valuable comments to improve our submission. We want to leave a gentle reminder due to the closing end time of the discussion period. We have tried to address all your concerns with detailed explanations and results and revised the paper correspondingly. We would really appreciate feedback to make sure the responses and revisions have addressed all your concerns, or whether there is a leftover concern we can address.
> >
> > Sincerely
> >
> > Authors of Paper2155

---

### Official Review · Reviewer_xvAD · 2022-10-22

**Confidence:** 4
**Correctness:** 3
**Technical Novelty And Significance:** 3
**Empirical Novelty And Significance:** 2
**Recommendation:** 6

**Clarity, Quality, Novelty And Reproducibility:**

Clarity and quality can be improved. The paper is novel and technically sound. Reproduction might be possible.

**Strength And Weaknesses:**

Pros:
(1)	The use of DRO for DGWD-ReID is technically sound.
(2)	Modified from the existing KL-DRO, the proposed Unit-DRO seems to work well for DGWD-ReID.
(3)	The extensive experiments show the effectiveness of the proposed method.

Cons:
(1) This paper claims a more general setting, i.e., DGWD-ReID. However, it's more of an easy-to-use approach, using only subject ID labels in DG-ReID. It can indeed be considered an improvement, but not a setting that is only suitable for real-world scenarios. Only the domain label and camera ID label don't seem to reveal so much location and environment information, which might be easier to know from the image background itself. Also, compared to the domain and camera ID, annotating subject ID seems to be much more expensive.

(2) This paper largely criticizes the demographic information (e.g., domain labels, camera IDs) used in existing DG-ReID methods. Even if the mentioned concerns (e.g., mainly privacy concerns) are true, we can also redefine demographic information without involving privacy concern. For example, we can apply clustering on all training sources, and assign the domain labels to different cluster centers (e.g., natural background or structured background). Training and the test evaluation might benefit if such obtained demographic information is also involved.

(3) As mentioned in the “Baseline Algorithm” of Section 3, similar to MetaBIN, the proposed method uses a mixture of batch normalization and instance normalization with learnable parameters. However, as discussed in MetaBIN, direct Batch-Instance Normalization may not generalize successfully on a given unseen target domain, so they use a meta-learning pipeline. How does the proposed method solve the problem without a meta-learning pipeline? Some discussion is necessary.

(4) As for the ablation study in Table 7, all of them are for different Unit-DRO components. How about the baseline without Unit-DRO and the baseline with KL-DRO (as Unit-DRO is modified from KL-DRO)?

(5) Some methods or labels in Table 5 and 6 are not defined. For example, in Table 5, what are A-IN, IBN, A-SN, IN stand for? In Table 6, what are the domains “P”, “A”, “C”, “S”, “V” and “L”? Besides, in Table 3, there are two MetaBINs. What is the difference?

(6) Minor points: in the abstract, “unweighted” might be “up-weighted”.


**Summary Of The Paper:**

This paper proposes a more general setting for person re-identification, i.e., domain generalizable person re-identification without demographics (DGWD-ReID). Based on the distributionally robust optimization (DRO) method for addressing the underlying uncertainty of domain distribution, it proposes a variant of DRO, named Unit-DRO, specifically for DGWD-ReID. Experimental results show the better performance of the proposed method compared to other baselines.

**Summary Of The Review:**

This paper proposes the Unit-DRO for DG-ReID without demographic information, which is novel and requires only subject ID labels. However, there are also some concerns and unclear points mentioned above. Therefore, I’m inclined to score “5”.

---

> ### Author Response · Authors · 2022-11-09
> **Response to Reviewer xvAD (1/3)**
>
> Thank you for the positive comments and careful reading, you have almost concluded the contributions of our work. We will answer the four concerns point by point.
>
> ----
>
> **[Concern 1]**. *Only the domain label and camera ID label don't seem to reveal so much location and environment information, which might be easier to know from the image background itself. Also, compared to the domain and camera ID, annotating subject ID seems to be much more expensive.*
>
> We detail the motivation from three aspects.
>
> 1. **Privacy Risks on using demographics in ReID tasks**. ReID is research about the person which is naturally with higher requirements for privacy [4]. Utilizing demographic information will bring the following two concerns.
> (1) Although the demographic data used in current ReID research is simple, e.g., the different campuses in Market1501 and CUHK, the demographic information in practical ReID deployments is much richer and can be obtained easily based on some inherent physical properties of cameras (such as network cameras’ MAC address and geographical locations). Thus, **the utilization of finer and richer physical properties of cameras will increase the leakage risk of the privacy information of pedestrians**. (2) The social relationships between different people are also possible to be exposed by demographic information, which may be more sensitive than geographical information. For example, if two-person IDs’s images are marked with the same camera, **their social relationships may be easily measured by counting their occurrence frequency in all cameras**. In summary, the uses of demographic data in real-world ReID applications have evident risks for personal privacy. Thus, we propose the setting of DGWD- ReID, where only a large-scale gallery of pedestrian images without any demographic data, e.g., camera IDs, can be used for training ReID models. Based on the setting of DGWD- ReID, researchers will be forced to **exploit invariant features from the training data itself, rather than resort to the side information**, i.e., demographic data, so as to reduce the risks of privacy leakage in ReID applications. We also agree that the annotation of subject ID is much more expensive and has more risks to personal privacy. We will tackle the challenging problem of unsupervised DG-ReID in future work. Thanks for your valuable suggestion again!
>
> 2. **The importance of DGWD setting in ML**. Learning generalizable models without domain labels is becoming an important matter of concern in the ML community [1][2]. Our algorithm can also be applied to generalized DG tasks where environment labels are unavailable at training time, either because they are difficult to obtain or due to privacy. In the following table, we compared [1,2] with Unit-DRO on the Waterbirds dataset, where Unit-DRO is superior to the baselines without the need for environment labels.
> | Method   | Test (worst group) |
> |----------|--------------------|
> | ERM      |               60.3 |
> | EIIL[2]     |               78.7 |
> | JTT[1]      |               86.7 |
> | Unit-DRO |               87.4 |
>
>
> 3. Finally, a previous study [3] shows that **how to optimally partition domains in training images that can benefit generalization capability most is still unclear**, which indicates the direct use of demographic data as domain labels may be inferior for learning domain-invariant representations. Thus, it is also a promising topic for DGWD-ReID as we discussed in the conclusion. In the fairness community, existing work has found that designing rational methodology can find domains that are maximally informative for downstream invariance learners. These domain IDs, and camera IDs make sense for humans, however, the similarities between different domains/camera views vary greatly. How to find more optimal domain partitions for ReID tasks is still an open problem.
>
> [1] EZ. Liu, et al.,Just train twice: Improving group robustness without training group information. ICML, 2021.
>
> [2] E Creager, et al., Environment inference for invariant learning. ICML, 2021.
>
> [3] M. Srivastava, et al., Robustness to Spurious Correlations via Human Annotations. ICML 2020.
>
> [4] W Zhuang, et al., Performance Optimization for Federated Person Re-identification via Benchmark Analysis, ACMMM, 2020.

---

> > ### Author Response · Authors · 2022-11-09
> > **Response to Reviewer xvAD (2/3)**
> >
> > **[Concern 2]** *This paper largely criticizes the demographic information (e.g., domain labels, camera IDs) used in existing DG-ReID methods. Even if the mentioned concerns (e.g., mainly privacy concerns) are true, we can also redefine demographic information without involving privacy concerns. For example, we can apply clustering on all training sources and assign the domain labels to different cluster centers (e.g., natural background or structured background).*
> >
> > Thank you for the valuable suggestion. In this work, demographic data refers to some self-owned information (e.g., camera/dataset IDs) that can be captured along with raw data collection. As to the pseudo domain labels learned by clustering, there have been some DG studies that try to automatically recover the original domain partition or even learn better domain partitions [1]. However, **the quality of inferred domain partitions is not good enough due to the imperfect clustering algorithms and some missing information e.g., the number of domains there should be**.
> >
> > To illustrate the point, we conduct simple experiments with our DG-ReID setting, where the K-means algorithm is applied for domain clustering. Even if we know the number of domains in advance, namely $K = 5$ i.e., the number of ReID datasets in the training stage, only $78.2%$ training instances can be clustered to the right domain. Specifically, if instances in cluster $i$ are mostly from domain $j$, we think instances in cluster $i$ are all classified to belong to $j$ and calculate the average accuracy over all domains.
> >
> > In this case, we adapt the evaluation protocol in Table.3 in the main paper and run the MetaBIN algorithm, **which can attain only 67.1% average mAP on the four test datasets with the inferred environments**.  In contrast, the performance is 72.3% that trained on the original dataset.
> >
> > However, as we discussed in future work, learning optimal domain partitions is still an open question and can be an interesting topic in future work. Thanks for your kind suggestion again.
> >
> > [1] Creager E, Jacobsen J H, Zemel R. Environment inference for invariant learning[C]//International Conference on Machine Learning. PMLR, 2021: 2189-2200.
> >
> > ----
> >
> > **[Concern 3]** *As for the ablation study in Table 7, all of them are for different Unit-DRO components. How about the baseline without Unit-DRO and the baseline with KL-DRO (as Unit-DRO is modified from KL-DRO)?*
> >
> > Thanks for the valuable suggestion, we will add more detailed ablation studies and make the following clarification to address your concerns.
> >
> > | Line | $\tau^*$       | $\|\mathcal{M}\|$ | R-1(%)      | mAP(%)      |
> > | ----  |----------------|-------------------|----------|----------|
> > |  1| KL-DRO      |              KL-DRO|     35.7 |     41.2 |
> > |  2| ERM     |               ERM|     57.8 |     66.2 |
> > |  3| 20       |               0|     63.6 |     71.5 |
> > |  4| [50,5,3]       |               800 |     64.2 |     72.1 |
> > |  5| [100, 5, 3]    |                 0 |     63.4 |     71.6 |
> > |  6| [100, 5, 3]    |               800 |       65 |     72.2 |
> > |  7| [100, 5, 3]    |              4000 |     63.9 |     71.9 |
> > |  8| **[100,20,5]** |           **800** | **65.4** | **72.8** |
> >
> > As shown in line 3, even with a constant $\tau^*$ and without a memory bank, the performance of Unit-DRO beats both ERM and KL-DRO by a large margin.
> >
> > To avoid grid search for hyper-parameters, we further propose an alternative approach (Linear-decay $\tau^*$), which models $\tau^*$ as a decreasing function of training steps. Specifically, $\tau^*=100(1-\frac{t}{T+1})$, such a method attains 65.0% R-1 accuracy and 72.3% mAP, which is comparable to the grid search result and simpler.
> >
> > ----
> >
> > **[Concern 4]** *How does the proposed method solve the problem without a meta-learning pipeline? Some discussion is necessary.*
> >
> > Thanks for your careful reading. The argument ``Directly using Batch-Instance Normalization (BIN) may not generalize successfully on a given unseen target domain’’ only means that BIN is inferior to MetaBIN, which uses the meta-learning pipeline. However, in our experiments, we found that using BIN directly without the meta-learning pipeline is also a strong baseline. The baseline ERM with BIN can attain $66.2%$ mAP. However, with the proposed Unit-DRO, the performance will be better than MetaBIN and needs no demographics, which again verifies the effectiveness of our method.

---

> > > ### Author Response · Authors · 2022-11-09
> > > **Response to Reviewer xvAD (3/3)**
> > >
> > > **[Concern 5]** *Some methods or labels in Table 5 and 6 are not defined. For example, in Table 5, what are A-IN, IBN, A-SN, IN stand for? In Table 6, what are the domains “P”, “A”, “C”, “S”, “V” and “L”? Besides, in Table 3, there are two MetaBINs. What is the difference?*
> > >
> > > ----
> > >
> > > *Q5.1 Some methods or labels in Table 5 and 6 are not defined.*
> > >
> > > Thanks for your careful reading. We have added descriptions on all these baselines or domains. For your convenience, we attach the descriptions as follows.
> > >
> > > In table.6, PACS includes four domains that are denoted by art (A), cartoons (C), photos (P), sketches (S) and VLCS also contains four domains that are denoted by Caltech101 (C), LabelMe (L), SUN09 (S), VOC2007 (V).
> > >
> > >
> > >  The baselines in Table.5 follow the settings in [2], where
> > > 1. A-IN: a naive model where we replace all the Batch Normalization(BN) layers in the Baseline by Instance Normalization(IN).
> > > 2. IBN: we add IN only to the last layers of Conv1 and Conv2 blocks of Baseline respectively.
> > > 3. A-SN: a model where we replace all the BN layers in the Baseline by Switchable Normalization (SN). SN [3] can be regarded as an adaptive ensemble version of normalization techniques of IN, BN, and LN (Layer Normalization).
> > > 4. IN: four IN layers are added after the first four convolutional blocks/stages of Baseline respectively.
> > >
> > > [1] Li, Da and Yang, Yongxin and Song, Yi-Zhe and Hospedales, Timothy M, Deeper, broader and artier domain generalization, ICCV, 2017
> > >
> > > [2] Jin X, Lan C, Zeng W, et al. Style normalization and restitution for generalizable person re-identification[C]//Proceedings of the IEEE/CVF Conference on Computer Vision and Pattern Recognition. 2020: 3143-3152.
> > >
> > > [3] Ping Luo, Jiamin Ren, Zhanglin Peng, Ruimao Zhang, and Jingu Li.  Differentiable learning-to-normalize via switch-able normalization.ICLR, 2019.
> > >
> > > ----
> > >
> > > *Q5.2  in Table 3, there are two MetaBINs. What is the difference?*
> > >
> > >
> > > Thanks for pointing out this problem. We observe that current DG ReID methods all apply a utopian model selection method to **report their best result by carefully checking the test performance after each training epoch**. So the numbers of training epochs corresponding to the best performance are varying for different test datasets We argue that such a model selection method is inadvisable. Under the DG setting, we should restrict access to the test domain data [1] for model selection. Thus, we simply use the last checkpoint and report its results as the final performance over all test datasets. The same methods in Table 3 denote the performance with two model selection methods respectively, where the higher one is the result with the exhaustive model selection using the test dataset, which is impractical. The lower one just chooses the last checkpoint, which is the same setting as ours. We also compare other methods with different model selection methods, where the results are shown in the following table and $^\dagger$ means the results of the last checkpoint are reported. We will update the table in the final submission for clarifying this point.
> > >
> > >
> > > | **Method**          | **R-1** | **mAP** |
> > > |---------------------|---------|---------|
> > > | DIR-ReID            |    63.8 |    71.2 |
> > > | DIR-ReID$^\dagger$  |    62.3 |    70.8 |
> > > | Group-DRO           |    57.1 |    65.9 |
> > > | Group-DRO$^\dagger$ |    56.7 |    65.6 |
> > > | MetaBIN             |    64.7 |    72.3 |
> > > | MetaBIN$^\dagger$   |    64.2 |    71.9 |
> > > | Unit-DRO$^\dagger$  |    65.4 |    72.8 |
> > >
> > > The results show that, without the utopian model selection method, there are always a certain degree of performance decline for existing DG ReID methods, which further indicates the advantages of the proposed Unit-DRO..
> > >
> > >
> > > [1] Gulrajani, Ishaan, and David Lopez-Paz. "In search of lost domain generalization." arXiv preprint arXiv:2007.01434 (2020).
> > >
> > > ----
> > >
> > > **[Concern 6]** *Minor points: in the abstract, “unweighted” might be “up-weighted”.*
> > >
> > > Thanks a lot for the valuable questions and careful reading and we have removed the typos.

---

> > > > ### Comment · Reviewer_xvAD · 2022-11-13
> > > > **Response to authors**
> > > >
> > > > Thank the authors for their detailed responses. Basically, most of my concerns have been addressed. I’d like to increase my score to “6”.  Hope these responses can be well reflected in the manuscript.

---

> > > > > ### Author Response · Authors · 2022-11-14
> > > > > **Summary of the paper revision**
> > > > >
> > > > > Dear Reviewer xvAD,
> > > > >
> > > > > Thanks again for your careful reading and valuable comments to improve our submission. The revision of our paper can be found in the following sections per your helpful suggestion.
> > > > >
> > > > >
> > > > > Section 4.2: we add A more detailed description of the experimental results, including methods for model selection, baseline in each table, and an introduction to symbols.
> > > > >
> > > > > Section 4.3: Detailed ablation study results and one alternative method for avoiding complex hyper-parameter searches.
> > > > >
> > > > > Appendix. A: We add more motivations for the proposed DGWD-ReID setting.
> > > > >
> > > > > Appendix. D: we add detailed information on the used datasets and baselines.
> > > > >
> > > > > ----
> > > > >
> > > > > Sincerely
> > > > >
> > > > > Authors of Paper 2155

---

### Official Review · Reviewer_2SbQ · 2022-10-25

**Confidence:** 4
**Correctness:** 4
**Technical Novelty And Significance:** 3
**Empirical Novelty And Significance:** 3
**Recommendation:** 6

**Clarity, Quality, Novelty And Reproducibility:**

The paper is well written, easy to follow and understanding. Codes are not available in the submission. The proposed method is novel to me.

**Strength And Weaknesses:**

**Strength**
1. The proposed DGWD-ReID setting is interesting, and the proposed method is novel.
2. The paper is easy to understand and well written.
3. Extensive experiments are conducted to demonstrate the superiority of the proposed method.

**Weakness**
1. In Line 7 of Algo. 1, does the model trained with Unit-DRO only use the cross-entropy loss function? Since in the last paragraph of Sec. 3, both cross-entropy and triplet losses are used as the baseline.
2.  Ablation studies in Tab. 7 are a little simple. It is unable to show a clear advantage of the proposed component by comparing results from two or three lines. Besides, how to select the optimal values for $\tau^{*}$ and |M|. I think they are either important or sensitive to the results.  As mentioned in Sec. 4.1, the scheme of multi-step $\tau$ seems complicated.  And in Tab. 7, it just simply compares the results with two options of M
3. In Tab. 3, there are two lines reporting the results of the same methods but with different numerical values


**Summary Of The Paper:**

This paper considers a problem of person re-identification in a more general setting, i.e., domain generalizable person ReID without demographics. To address this, it introduces distributionally robust optimization (DRO) to learn robust person ReID models that perform well on all possible data distributions within the uncertainty set without demographics. Furthermore, it analyzes and reformulates the popular KL-DRO by applying the change-of-measure technique, and then propose a simple yet efficient approach, Unit-DRO. Extensive experiments are conducted on several ReID tasks to show the superiority of the proposed method.

**Summary Of The Review:**

This paper proposes a novel Unit-DRO approach to tackle the task of domain generalizable person ReID without demographics. Although the proposed method is novel, and achieves superior performance on both person ReID and general DG tasks. there are some limitations as mentioned in the weakness. So I give the score of 5 and would like to hear the feedback from the author.

---

> ### Author Response · Authors · 2022-11-09
> **Response to Reviewer 2SbQ**
>
> Thank you for the careful reading and valuable feedback on our submission and hope our response could solve your concerns.
>
> ----
>
> **[Concern 1]**. *In Line 7 of Algo. 1, does the model trained with Unit-DRO only use the cross-entropy loss function? Since in the last paragraph of Sec. 3, both cross-entropy and triplet losses are used as the baseline.*
>
> Thanks a lot for pointing out this misleading content. Actually, the model is trained by both the cross-entropy loss and the triplet loss, namely the $\ell$ in Algorithm 1 is the summation of cross-entropy loss and the triplet loss, which has been widely adopted in ReID studies. We make it clear in the revision.
>
> ----
>
> **[Concern 2]**. *Ablation studies in Tab. 7 are a little simple.*
>
> Thanks for the valuable suggestion, we will add more detailed ablation studies and make the following clarification to address your concerns.
>
> | Line | $\tau^*$       | $\|\mathcal{M}\|$ | R-1      | mAP      |
> | ----  |----------------|-------------------|----------|----------|
> |  1| KL-DRO      |              KL-DRO|     35.7 |     41.2 |
> |  2| ERM     |               ERM|     57.8 |     66.2 |
> |  3| 20       |               0|     63.6 |     71.5 |
> |  4| [50,5,3]       |               800 |     64.2 |     72.1 |
> |  5| [100, 5, 3]    |                 0 |     63.4 |     71.6 |
> |  6| [100, 5, 3]    |               800 |       65 |     72.2 |
> |  7| [100, 5, 3]    |              4000 |     63.9 |     71.9 |
> |  8| **[100,20,5]** |           **800** | **65.4** | **72.8** |
>
> 1. **Choosing the best** $|\mathcal{M}|$. $|\mathcal{M}|$ denotes the size of memory. $|\mathcal{M}|=0$ means that we use the batch data as the normalizer $\mathbb{E}_\mathcal{P}[e^{\ell(x,y;\theta)/\tau^*}]$, which cannot attain the best performance (line 3,5 in the table). A huge $\mathcal{M}$ contains much out-of-date data and also attains inferior results (line 7). In our experiments, we propose to use $|\mathcal{M}|=10\times bsz$, where $bsz$ is the batch size during training.
>
> 2. **Choosing the best** $\tau^*$.  Actually, the selection of multi-step $\tau^*$ is not complicated. At the first few epochs, we simply set $\tau^*$ to a large value such as $100$. In epochs 40, and 70, we decay it to smaller values. When $|\mathcal{M}| = 800$, the performance gap will not be much sensitive to different choices of $\tau^*$.
>
> 3. **Sensitivity**. As shown in line 3, even with a constant $\tau^*$ and without a memory bank, the performance of Unit-DRO beats both ERM and KL-DRO by a large margin, which is not too sensitive.
>
> To avoid grid search for hyper-parameters, we further propose an alternative approach (**Linear-decay** $\tau^*$), which models $\tau^*$ as a decreasing function of training steps. Specifically, $\tau^*=100(1-\frac{t}{T+1})$, such a method attains $65.0$ R-1 accuracy and $72.3$ mAP in the setting of Table.3, which is comparable to the grid search result and simpler.
>
> ----
>
> **[Concern 3]**. *The same methods but with different numerical values in Table .3.*
>
> Thanks for pointing out this problem. We observe that current DG ReID methods all apply a utopian model selection method to **report their best result by carefully checking the test performance after each training epoch**. So the numbers of training epochs corresponding to the best performance are varying for different test datasets. We argue that such a model selection method is inadvisable. Under the DG setting, we should restrict access to the test domain data [1] for model selection. Thus, we simply use the last checkpoint and report its results as the final performance over all test datasets. The same methods in Table 3 denote the performance with two model selection methods respectively, where the higher one is the result with the exhaustive model selection using the test dataset, which is impractical. The lower one just chooses the last checkpoint, which is the same setting as ours. We also compare other methods with different model selection methods, where the results are shown in the following table and $^\dagger$ means the results of the last checkpoint are reported. We will update the table in the final submission for clarifying this point.
>
>
> | **Method**          | **R-1** | **mAP** |
> |---------------------|---------|---------|
> | DIR-ReID            |    63.8 |    71.2 |
> | DIR-ReID$^\dagger$  |    62.3 |    70.8 |
> | Group-DRO           |    57.1 |    65.9 |
> | Group-DRO$^\dagger$ |    56.7 |    65.6 |
> | MetaBIN             |    64.7 |    72.3 |
> | MetaBIN$^\dagger$   |    64.2 |    71.9 |
> | Unit-DRO$^\dagger$  |    65.4 |    72.8 |
>
> The results show that, without the utopian model selection method, there is always a certain degree of performance decline for existing DG ReID methods, which further indicates the advantages of the proposed Unit-DRO.
>
>
> [1] Gulrajani, Ishaan, and David Lopez-Paz. "In search of lost domain generalization." arXiv preprint arXiv:2007.01434 (2020).

---

> > ### Author Response · Authors · 2022-11-14
> > **A Gentle Reminder of Feedbacks**
> >
> > Dear Reviewer 2SbQ,
> >
> > Thanks again for your careful reading and valuable comments to improve our submission. Reveiwer xvAD who has similar questions to you has had a discussion with us and most of his concerns are addressed. We want to leave a gentle reminder due to the closing end time of the discussion period. We have tried to address all your concerns with detailed explanations and results and revised the paper correspondingly. We would really appreciate feedback to make sure the responses and revisions have addressed all your concerns, or whether there is a leftover concern we can address.
> >
> > Sincerely
> >
> > Authors of Paper2155

---

### Decision · Program_Chairs · 2023-01-20

**Decision:**

Reject

**Justification For Why Not Higher Score:**

N/A

**Justification For Why Not Lower Score:**

N/A

**Metareview: Summary, Strengths And Weaknesses:**

This paper proposes a more general setting for person re-identification, i.e., domain generalizable person re-identification without demographics (DGWD-ReID). Based on the distributionally robust optimization (DRO) method for addressing the underlying uncertainty of domain distribution, authors proposed a variant of DRO, named Unit-DRO, specifically for DGWD-ReID tasks and empirically showed better performance of the proposed method compared to other baselines. While reviewers generally think the new setting is interesting, neither can give a strong support of acceptance and they all point out various concerns of limitations such as lack of convincing ablation studies where some results seem negative without clear justification,  lack of comparison with stronger baselines in the new setting (as the authors claim it is novel but it seems possible to adapt some existing works for a more convincing baselines), and lack of thorough empirical analysis and discussion of pros and cons of the proposed method, and finally there is a lack of discussions with more recent related work. On a side note, this work seems to be a resubmission of ICLR2022, but there seems very little improvement as compared with the previous rejected paper, e.g., there is no any comparison or discussion with recent works and most cited papers are before 2021. Overall, I think this is an OK paper with incremental contributions, but it is not significant enough for acceptance.

**Summary Of Ac-Reviewer Meeting:**

N/A